



# Thermo-mechanical numerical modelling of the South American subduction zone: a multi-parametric investigation

Vincent Strak[1] and Wouter P. Schellart[1]

[1]Department of Earth Sciences, Vrije Universiteit Amsterdam, Amsterdam, 1081HV, Netherlands

*Correspondence to*: Vincent Strak (v.strak@vu.nl)

**Abstract.** The South American subduction zone remains a topic of debate with long-lasting questions involving the origin of non-collisional orogeny and the effect of very large trench-parallel extent, slab sinking to great mantle depths, and aseismic ridge subduction. A key to help solve those issues is through studying the subduction zone dynamics with buoyancy-driven numerical modelling that uses constrained independent variables in order to best approximate the dynamics of the real

subduction system. We conduct a parametric investigation on the effect of upper mantle rheology (Newtonian or non-Newtonian), subduction interface yield stress and slab thermal weakening. As a means of constraining those model variables we attempt to find best-fits by comparing our model outcomes with the present-day upper- and lower-mantle slab geometry observed on tomography models and obtained from earthquake hypocentre locations, as well as with estimates of Cenozoic velocities obtained from kinematic reconstruction. Key ingredients that need to be reproduced are slab flattening close to the

surface, strong oscillation of the Farallon-Nazca subducting plate velocity and progressive decrease in trench retreat rate after a long period of time. We include these ingredients to define a model fitting score that contains a total of 9 criteria. Our best fitting model involves significant slab thermal weakening in order to attain the fast Farallon-Nazca subducting plate velocity and to better reproduce the subduction partitioning in the past 48 Myr, due to strong reduction of the shear stresses resisting downdip slab sinking and of the slab bending resistance. We further find that a non-Newtonian upper mantle rheology promotes

slab folding and realistic associated oscillation of the subducting plate velocity. Our parametric study also indicates that the subduction interface must be weak in agreement with earlier laboratory subduction models, but not too weak, with a yield stress of ~14–21 MPa, otherwise the fit becomes poor. Our models moreover suggest that slab folding at the 660 km discontinuity can be a cause of the Farallon-Nazca subducting plate velocity oscillation. Whether and how this slab folding process induces periodic/episodic variations in deformation of the Andes remains an open question that requires further

research.

## 1. Introduction

The South American subduction zone challenges our view of subduction dynamics and mountain building because it has enigmatically produced the Andes as a result of oceanic lithosphere subduction. Thus, the orogeny was not formed from the classical scenario of continental collision. It is also an intriguing subduction system with the widest along-trench dimension





on Earth and subduction of aseismic ridges, two factors that likely impact on subduction dynamics and on the resulting surficial velocities and natural hazards. Building geodynamic models is key to improve our understanding of the complex South American subduction system. Running 3-D geodynamic subduction models would be an ideal approach to study the subduction zone dynamics. With the continued improvement in computational power (Jadamec, 2016) the development of 3-D geodynamic modelling becomes indeed more realistic. However, despite the powerful computational resources, running 3-D

models remains an expensive task as the models can require months or years to complete (e.g. Schellart, 2017), in particular for such a large setting as the South American subduction zone. Therefore, we propose with this study to use 2-D models to do a parametric investigation of several independent physical parameters in order to test their effect on the subduction process and to calibrate these parameters for future modelling in 3D space. The two-dimensional approach is suited for this study since 2-D models well reproduce subduction dynamics at the central section of wide subduction zones.

The objectives of this paper are twofold: (1) calibrate independent variables for use in future 3-D modelling by comparing model outcomes with a range of geophysical and kinematic data, and (2) parametrically investigate the effect of the changed independent variables to get generic quantitative insights into how they affect subduction dynamics. The tested variables are upper mantle rheology (Newtonian or non-Newtonian), subduction interface yield stress and slab thermal weakening.

The role of a Newtonian or non-Newtonian upper mantle was previously investigated in numerical convection and subduction models (Christensen and Yuen, 1989; King and Hager, 1990; van den Berg et al., 1993; Jadamec and Billen, 2010; Billen and Jadamec, 2012; Holt and Becker, 2017). Convection modelling studies show that models using a power-law viscosity produce faster average convection rates as well as higher amplitudes of temporal velocity variations (Christensen and Yuen, 1989; van den Berg et al., 1993). Additionally, instantaneous subduction models demonstrate that using a power-

law viscosity in the upper mantle leads to faster slab sinking rates with enhanced localisation of high mantle velocities in regions of high strain rates (Jadamec and Billen, 2010; Billen and Jadamec, 2012). Furthermore, Holt and Becker (2017) used 2-D and 3-D time-evolving subduction models to show that a power-law rheology in the mantle produces slower trench retreat rates. They propose that this is owed to the mantle shear force, which is the dominant factor resisting downdip slab motion, being reduced to a greater extent than the pressure gradient across the slab, which is the dominant factor resisting slab rollback.

These latest results point to the fact that the effects of mantle rheology on partitioning between subducting plate motion and slab rollback motion should be examined further.

Another parameter that was previously recognised to provide a control on subduction dynamics is the subduction interface strength (Čížková and Bina, 2013; Duarte et al., 2013; Holt et al., 2015). Čížková and Bina (2013) show that the subducting plate velocity increases with decreasing subduction interface strength (viscosity). Similarly, slab rollback rate is

increased, although more moderately, resulting in a decrease of the slab dip angle. Higher slab dip angles and reduced subducting plate velocities at higher subduction interface strengths are further shown to increase the period of slab folding at the 660 km discontinuity. Moreover, models with stress-dependent weakening of the subduction interface from Holt et al. (2015) agree with these results but indicate a stronger impact on increasing trench retreat rate, which also decreases the slab



dip angle. The models however show that after interacting with the upper-lower mantle discontinuity the slab rolls back at a
slower rate than the Newtonian reference model but the subducting plate velocity is significantly faster, which thus leads to an
increase of the slab dip angle in the upper mantle. In addition to affecting slab dip angle and subducting plate and trench retreat
velocities, the models from Holt et al. (2015) with reduced subduction interface strength produce the greatest overriding plate
compression they observe, including compression in the forearc during the free-sinking phase. This is opposite to laboratory
subduction models, in which a weak plastic subduction interface produces forearc and backarc extension (Duarte et al., 2013;
Meyer and Schellart, 2013) and a stronger interface induces forearc shortening (Duarte et al., 2013). In more recent laboratory
work using an advanced imaging technique to quantify overriding plate deformation, a weak interface is shown to produce a
zone of forearc extension, a zone of forearc shortening, and overall backarc extension (Chen et al., 2016). We note, however,
that the two-dimensional model setup in Holt et al. (2015) is diametrically different from the 3-D analogue models above,
which may explain the difference in how subduction interface strength affects overriding plate deformation.

Slab thermal weakening has been implemented in some numerical subduction modelling studies (van Hunen and Allen,
2011; Čížková and Bina, 2013) but its impact on subduction dynamics has not yet been evaluated. van Hunen and Allen (2011)
used a temperature-dependent viscosity in order to facilitate slab breakoff by reducing the effective viscosity of the
progressively warmed sinking slab. Čížková and Bina (2013) imposed temperature-dependent viscosity on young, thus initially
weak, and old, thus initially strong, slabs. Their models show that the weak slabs deform by horizontal buckling at the 660-
km discontinuity whereas the strong slabs do not buckle and remain flat. However, the effect on slab deformation in their
models could mostly be due to slab age control on initial slab thickness and temperature, and it is impossible to extract a causal
relationship with temperature-dependent viscosity. Thus, the impact of slab thermal weakening through temperature-dependent
slab viscosity remains to be tested parametrically.

In order to find best-fitting values for the tested independent variables we compare with the natural prototype a range
of dependent variables including slab geometry and surficial velocities using tomography, seismology and a kinematic
reconstruction study. We also compare overriding plate deformation to ensure consistency. The model-nature comparison
indicates that to best replicate slab geometry and surficial kinematics the model requires the use of a non-Newtonian upper
mantle, a weak subduction interface with a yield stress of ~14–21 MPa, and significant slab thermal weakening.

## 2. Methods

### 2.1. Approach and governing equations

The regional models were designed to conduct a parametric investigation on the effect of upper mantle rheology
(linearly or non-linearly viscous), subduction interface yield stress $\sigma_y$ and slab thermal weakening on South American
subduction dynamics over a long timescale (~60-200 Myr) and large spatial dimensions (11600 km laterally and 2900 km
vertically).





To study the effect of the investigated parameters on the dynamics of South American subduction we present numerical buoyancy-driven subduction modelling using version 2.7 of the code *Underworld 2* (Mansour et al., 2019). The code is based on *Underworld* (Moresi et al., 2003, 2007) and uses Python scripts to build the model setup (mesh dimension and refinement, model geometry, rheological functions) and control the solver types and options. The models are two dimensional and use a Cartesian geometry (**Fig. 1**). The 2-D approach is a reasonable approximation considering that we simulate the subduction

process at the centre of a very wide subduction system where toroidal mantle flow is minimal and slab and plate kinematics is very similar as in 3-D subduction models at the centre of the subduction system. The Cartesian reference frame may be a more important limitation as using an annulus sector would better approximate the spherical geometry of the Earth. However, this implementation was not yet stable in the version of the code (*Underworld* 2.7) we used here. 14 models were run on Dutch and Australian national supercomputers using 48 cores. The models ran for ~60 hours to complete 6000–7000 time steps.

To simulate layers of different composition (density and viscosity) in the lithosphere and sub-lithospheric mantle the models use sets of Lagrangian particles within a standard Eulerian finite element mesh (Moresi et al., 2003). The Lagrangian particles can move relative to the mesh and both the particles and mesh are used for discretisation in order to solve the governing equations (pressure, temperature, and velocity are solved on the mesh). The numerical subduction models are driven by temperature-dependent density contrasts and therefore use an internal approach as described in Schellart and Strak (2016)

in order to ensure that all forces driving and resisting subduction are enclosed within an isolated system. The code solves reduced forms of the Navier-Stokes equations for an incompressible Stokes fluid at very low Reynolds number (inertial forces are negligible). The models use the Boussinesq approximation, which assumes that variations in density have no effect on the flow field except for the initial density distribution that generates the driving body force. Moreover, assuming incompressibility means that the rate of change of density due to thermal warming/cooling, and thus the rate of thermal expansion and thermal

contraction, is very small with respect to the velocity gradients. It can thus be neglected and the equation for conservation of mass simplifies to the following incompressible form of the continuity equation:

$$\nabla \cdot \boldsymbol{u} = 0 \tag{1}$$

where $\boldsymbol{u}$ is the velocity vector. It also follows that the momentum conservation equation reduces to:

$$\nabla P - \nabla \cdot \boldsymbol{\tau} = \Delta\rho \cdot \boldsymbol{g} \tag{2}$$

where $P$ is the dynamic pressure, $\boldsymbol{\tau}$ is the deviatoric component of the stress tensor, $\rho$ is the density with $\Delta\rho = \rho - \rho_{ref}$ and $\boldsymbol{g}$ is the gravitational acceleration. The density variations are controlled by temperature following the equation of state:

$$\rho(T) = \rho_{ref}\big[1 - \alpha(T - T_{ref})\big] \tag{3}$$

which, considering density contrasts, simplifies to:

$$\Delta\rho(T) = \rho_{ref}\alpha(T - T_{ref}) \tag{4}$$

with $T$ temperature, α the thermal expansivity, and $\rho_{ref}$ the reference upper mantle density of 3230 kg m$^{-3}$ in the natural prototype at temperature $T_{ref}$ of 1300 °C (Cloos, 1993). We chose a value of 1.5 10$^{-5}$ K$^{-1}$ for α, which represents an intermediate value for the whole mantle estimated to vary between 1.0 10$^{-5}$ K$^{-1}$ and 3.5 10$^{-5}$ K$^{-1}$ (Steinberger and Calderwood, 2006) and is





similar as in previous numerical subduction models (e.g. Holt et al., 2015). This intermediate value allows us to simulate a

scaled density of ~3292.985 kg m$^{-3}$ for the oceanic lithosphere, thereby giving a subducting plate-upper mantle density contrast $\Delta\rho(T)_{SP-UM}$ of -62.985 kg m$^{-3}$. In order to include the effects of compositional buoyancy, we also define the following composition-dependent density contrast, which needs to exclude the effects of temperature previously considered:

$$\Delta\rho(C) = \Delta\rho(C,T) - \Delta\rho(T) \tag{5}$$

We then impose $\Delta\rho(C,T)_{OP-UM} = 0$ for the overriding plate layers in order to keep the overriding plate neutrally buoyant. This allows to reduce the number of possible causes of deformation to the shear stresses since we neglect the gravitational

potential energy in the overriding plate. Furthermore, thermal advection and diffusion are solved using the energy conservation equation:

$$\frac{\partial T}{\partial t} + \boldsymbol{u} \cdot \nabla T = \kappa \nabla^2 T \tag{6}$$

where $t$ is time and $\kappa$ is the thermal diffusivity. The Lagrangian markers track lithology, corresponding properties and stress histories, as they are advected according to the velocity field and leave the Eulerian grid undeformed. Information carried by the advected Lagrangian particles allows to link strain rate and stress to the effective viscosity using the constitutive

relationship:

$$\boldsymbol{\tau} = 2\eta\dot{\boldsymbol{\varepsilon}} \tag{7}$$

where $\eta$ is the dynamic viscosity and $\dot{\boldsymbol{\varepsilon}}$ the strain rate tensor expressed as:

$$\dot{\boldsymbol{\varepsilon}} = 1/2(\nabla \cdot \boldsymbol{u} + \nabla \cdot \boldsymbol{u}^{\mathrm{T}}) \tag{8}$$

## 2.2. Scaling

Non-dimensionalisation is achieved using the following scaling factors (indicated by s subscript) such that $X = \bar{X}X_s$ where the upper bar indicates dimensionless quantity whereas nothing is used to indicate scaled value and $X$ represents any

possible parameter with $\sigma$ for stress (see **Table 1** for values of the characteristic scaling parameters):

$$l_s = h, \eta_s = \eta_{ref}, T_s = \Delta T, t_s = \frac{l_s^2}{\kappa}, u_s = \frac{\kappa}{l_s}, \sigma_s = \frac{\kappa\eta_s}{l_s^2}, \nabla_s = \frac{1}{l_s} \tag{9}$$

Using these scaling factors and equations 4 and 5 we can thus substitute the scaled parameters for their dimensionless counterparts in equation 2, which after rearranging leads us to write the following dimensionless momentum conservation equation, similar as Sandiford and Moresi (2019):

$$\bar{\nabla}\bar{P} - \bar{\nabla} \cdot \bar{\boldsymbol{\tau}} = Ra(\bar{T} - 1) + Rb \tag{10}$$

where $Ra = \frac{l_s^3 \rho_{ref} g\alpha\Delta T}{\kappa\eta_s}$ is the Rayleigh number, which has a value of $4.3 \times 10^7$ in all models, and $Rb = \frac{l_s^3 g\Delta\rho(C)}{\kappa\eta_s}$ is the

compositional Rayleigh number (van Hunen et al., 2002). Similarly, using the scaling factors and substituting the scaled parameters for their dimensionless counterparts in equation 6 leads to the following dimensionless energy conservation equation:





$$\frac{\partial \overline{T}}{\partial \overline{t}} + \overline{\boldsymbol{u}} \cdot \overline{\nabla} \overline{T} = \overline{\nabla}^2 \overline{T} \qquad (11)$$

## 2.3. Model setup and rheology

The models build on earlier modelling work from Schellart (2017) by implementing thermal advection and diffusion.
The reference model (see **Table 2** and **Fig. 1**) has a length of 11600 km and a height of 2900 km to include the lithospheric plates and both upper and lower mantle reservoirs. The subducting oceanic lithosphere represents the Nazca-Farallon plate and is 5700 km long including a 200-km-long initial slab perturbation that dips at an angle of 29° in order to initiate the subduction process. The subducting plate is composed of three layers, a top, a core and a bottom layer of 30, 20 and 30 km thick, respectively (**Fig. 1a**). The overriding plate simulates the South American plate and contains a top and a bottom layer. The
overriding plate top layer has a uniform thickness of 30 km to simulate the crust while the overriding plate bottom layer is made up of four regions with different lithospheric mantle thicknesses: a 30-km-thick region extending up to 1300 km away from the trench that includes a forearc and backarc domain, a transitional region that linearly increases in thickness over 200 km to simulate the transition from backarc domain to cratonic domain, a 120-km-thick region that extends between 1500 and 4500 km from the trench to simulate a ~150-km-thick South American cratonic lithosphere (Heit et al., 2007), and a transitional
region that linearly decreases in thickness over 200 km to simulate the Brazilian passive margin. The overriding plate layers are then linearly tapered over 1500 km of the trailing edge to simulate the western segment of the Atlantic oceanic lithosphere that progressively decreases in thickness up to the Atlantic mid oceanic ridge. The forearc and backarc thicknesses were determined following earlier work suggesting that their presence is prevalent in ocean-continent subduction zones of the Pacific domain (Curie and Hyndman, 2006).
We implemented a simplified temperature distribution by ascribing a unique initial value in the plates and in the whole mantle. This method allows to drive subduction by thermal buoyancy neglecting the effects of vertical mantle temperature and density variation (i.e. adiabatic gradient, progressive density increase with depth) but considering thermal diffusion and the associated density changes occurring in, and in the vicinity of, the slab. With this approach we chose to minimise the complexity of the independent variables in order to get a better control on how they affect the dependent variables, therefore
facilitating the analysis and interpretation of the model outcomes.

All materials are treated as viscous except for the upper layer of the subducting plate, which has a visco-plastic rheology to allow for sufficient reduction of the effective viscosity, thereby ensuring continued single-sided subduction (Crameri et al., 2012). This simplified rheological approach has proven successful in producing consistent subduction dynamics as shown extensively in previous modelling research (King and Hager, 1990; Stegman et al., 2006; Capitanio et al., 2007; Schellart et
al., 2007; van Hunen and Allen, 2011; Schellart and Moresi, 2013; Schellart, 2017). Apart from the subducting plate top layer, all other plate layers and the lower mantle are Newtonian whereas for the upper mantle both a Newtonian and power-law rheology were tested (**Table 2**). In nature, mantle rocks deform by dislocation and diffusion creep following an Arrhenius flow





law with temperature and pressure dependence (Karato and Wu, 1993; Hirth and Kohlstedt, 2003). For dislocation creep, we computed a dynamic effective viscosity following the Arrhenius flow law but neglecting the effect of pressure (van Keken et

al., 2008), which thus writes:

$$\eta = \frac{1}{2} A^{-1/n} \exp\left(\frac{E}{nRT}\right) \dot{\varepsilon}_{II}^{(1-n)/n} \tag{12}$$

where $\eta$ is the dynamic effective viscosity, $A$ is the pre-exponential factor, $n$ is the power-law exponent, $E$ is the activation energy, $R$ is the universal gas constant, and $\dot{\varepsilon}_{II}$ is the second invariant of the strain rate tensor. For the lower mantle we assume a diffusion creep rheology and we choose a value of dynamic effective viscosity that it is 100 times larger than the upper mantle reference viscosity in the reference model UMnl. The upper mantle viscosity is restricted to the dimensionless range

of 0.1–1, scaling to 3.5 $10^{19}$–3.5 $10^{20}$ Pa s, by imposing cut-off values. Dimensionless viscosity in the lithospheric plates is 1000, 1000 and 50 in the subducting plate top, core and bottom layer, respectively (**Fig. 1b** and **Table 1**). The lower viscosity of the subducting plate bottom layer allows to simulate the weak rheology of a lithospheric mantle composed of harzburgite. The viscosity of the subducting plate top layer sinking below 200 km depth is significantly decreased from 1000 to 50 assuming transformation into eclogite, which experimentally deforms according to a similar flow law as harzburgite (Jin et al., 2001).

Plasticity in the 30-km-thick subducting plate top layer is simulated using a yield stress of 21 MPa in the reference model, above which the effective viscosity is reduced following a von Mises yield criterion as follows:

$$\eta = \frac{1}{2} \frac{\sigma_y}{\dot{\varepsilon}_{II}} \tag{13}$$

We limit the plastic weakening of the subducting plate top layer so that it cannot decrease to less than 0.1 times the upper mantle reference viscosity. The values we use for the viscosity structure agree with previous modelling studies, notably regarding the slab to upper mantle viscosity ratio (Funiciello et al., 2008; Schellart, 2008; Ribe, 2010; Rolf et al., 2018) and

the lower to upper mantle viscosity ratio (Steinberger and Calderwood, 2006). In some models, thermal weakening of the slab is simulated using a non-dimensional temperature-dependent viscosity that follows a linearized Arrhenius flow law (Ratcliff and Schubert, 1996; Zhong et al., 2000):

$$\eta = exp(E' \cdot (-\bar{T})) \tag{14}$$

where $E'$ is the dimensionless activation energy coefficient and is generally set to ~7 or ~9 (Zhong et al., 2000).

        The number of mesh elements is 1024 horizontally and 512 vertically using 20 Lagrangian particles per cell. A mesh

refinement was implemented above 150 km depth and between 3800 and 5700 km horizontally, leading to a resolution of 3 km vertically and 5 km horizontally in the refined area, but ~6 km vertically and ~15 km horizontally in the non-refined domain. Velocity boundary conditions are rigid free-slip for all four box walls. This allows to minimise boundary effects on the flow occurring in the system (Enns et al., 2005; Schellart and Moresi, 2013) in addition with using a very wide spatial domain (Schellart and Moresi, 2013; Garel et al., 2014). Thermal boundary conditions are isothermal at the top (0 °C) and

bottom ($T_{ref}$) and insulating at the side walls with zero heat flux for all four boundaries.

## 2.4. Tested independent variables



In this study we parametrically investigate the effect of the following variables considering model UMnl-ref as the reference model (**Table 2**). Models UMlin, UMnl-A1e6, UMnl-A1.25e7 and UMnl-ref ($A = 3 \times 10^6$ Pa$^n$ s) tested the effect of using a Newtonian or non-Newtonian upper mantle rheology with additionally varying the intensity of the non-linearity by

changing the pre-exponential factor $A$ in equation 12. Models UMnl-yield7, UMnl-yield14, UMnl-yield28, UMnl-yield35 and UMnl-ref ($\sigma_y = 21$ MPa) tested the effect of changing the subduction interface yield stress $\sigma_y$. Furthermore, models UMnl-E'3, UMnl-E'5, UMnl-E'7, UMnl-E'9, UMnl-E'11 and UMnl-E'13 tested the impact of slab thermal weakening with changing the dimensionless activation energy $E'$ in equation 14.

### 2.5. Dependent variables compared with natural case

To investigate subduction dynamics, we tracked the spatial and temporal variations of several dependent variables of the model, namely the slab dip angle ($\alpha_{dip}$) at three depth ranges (60–120, 300–600 and 1000–1800 km), surficial velocities of the trench ($V_T$), subducting plate ($V_{SP}$), overriding plate trailing edge and craton-back-arc transition zone ($V_{OP\_te}$ and $V_{OP\_tz}$, respectively), and deformation rate produced in the forearc and backarc domain of the overriding plate (obtained from calculating $V_T$-$V_{OP\_tz}$), as well as horizontal deviatoric normal stress in the overriding plate ($\sigma_{XX}$) and horizontal deviatoric

shear stress in the mantle just below the overriding plate ($\sigma_{YX}$). In order to perform our model-nature comparison, we compared the slab dip angle measured at the three depth ranges and the surficial velocities with their natural counterpart estimated using tomography, seismology and a kinematic reconstruction study. We also compared overriding plate deformation, although only to ensure consistency since overriding plate viscosity was not varied, whereas it provides a first-order control on the magnitude of deformation.

Tomography models (Amaru, 2007; Simmons et al., 2012; Lu et al., 2019) and the location of earthquake hypocentres (Hayes et al., 2018) indicate that the slab of the South American subduction zone dips eastward in the whole mantle (**Fig. 2**). They also reveal that the slab dip angle between -33˚ and 10˚ latitude varies with depth with estimated values ranging between ~0˚ and 25˚ at 60–120 km depth, ~41˚ and 75˚ at 300–600 km depth, and 53˚ and 62˚ at 1000–1800 km depth (**Fig. 2a,d**). Notably, two flat slab segments, with a dip angle not exceeding 10˚ as defined by Schellart (2020), are observed between -2.5˚

and -16˚ latitude and between -24˚ and -32.5˚ latitude (**Fig. 2a**). They both occur at a depth of 75–100 km (**Fig. 2d**) and have a length of ~300 km (**Fig. 2b,c**). Furthermore, the high-velocity anomaly observed in the lower mantle is thicker than in the upper mantle (**Fig. 2e,g**), suggesting some thickening mechanism like folding of the slab (e.g. Ribe et al., 2007) or the signal may be broader due to thermal diffusion and/or reduced tomographic resolution (Shephard et al., 2017). We note that some tomography models show a disconnection of the high-density anomaly in the lower mantle below 1500 km (**Fig. 2e**) while

other models image a continuous slab down to 2200 km and even further (**Fig. 2f,g**). Concerning surficial velocities, kinematic reconstruction studies can be used as a means to provide estimates. Using the model of Matthews et al. (2016) in an Indo-Atlantic hotspot reference frame we calculate the subducting plate velocity ($V_{SP}$) in the past ~48 Myr with the motion of the oldest isochron possible to track on the Nazca plate (Müller et al., 2016). We also give an approximation of the trench retreat velocity ($V_T$), ignoring the component due to overriding plate deformation, by calculating the displacement of the South



American plate over the past 120 Myr (**Fig. 3a**). $V_{SP}$ varies significantly over time since 48 Ma and, except for a relatively low value of ~3 cm/yr at ~45–48 Ma, it fluctuates between ~6 and ~10 cm/yr in the past ~45 Myr to reach a value of ~6 cm/yr at present. The time evolution of $V_T$ shows some fluctuation and an overall reduction from ~4.2 cm/yr at 120 Ma to ~1.4 cm/yr at present, with a very gentle decrease from ~2.5 cm/yr to ~1.4 cm/yr in the past 48 Myr. The partitioning of subduction into subducting plate and slab rollback motion can furthermore be expressed as $V_{SP}/(V_{SP}+V_T)$, which we refer to as subduction

partitioning ratio, and it can be calculated for the past ~48 Myr (**Fig. 3b**). The subduction partitioning ratio shows a stepwise increasing trend with lower and upper values of 0.57 and 0.85, meaning that convergence is progressively accommodated more by subducting plate trenchward motion and less by trench retreat. As to deformation in the South American overriding plate, tectonic reconstructions and balanced cross-section studies (Kley and Monaldi, 1998; Arriagada et al., 2008; Schepers et al., 2017) provide an estimate of ~240–410 km cumulative shortening generated in the forearc and backarc domain since the mid-

Eocene (~45 Ma). Furthermore, a study focusing on basin migration and balanced cross-sections (McQuarrie et al., 2005) provides an estimate of 530–580 km of shortening since the late Cretaceous (~70 Ma). All together, these studies suggest a mean shortening rate of ~5.33–9.11 mm/yr. Replicating in the present study these distinct slab geometry and surficial velocities using internally-driven geodynamic modelling provides a way to find a range of reasonable values that can be used for the independent variables in future 3-D models of the South American subduction zone.

## 3. Results

### 3.1. General model dynamics and effects of upper mantle rheology

The subduction evolution in the reference model occurs in two phases. The first phase corresponds to the slab sinking freely into the upper mantle until it reaches the 660 km discontinuity, which we refer to as upper mantle subduction phase. During this period, the slab sinks rapidly to attain a quasi-linear geometry and a high dip angle before interacting with the

upper-lower mantle discontinuity after ~25 Myr in the reference model (**Fig. 4a,b and Fig. 5a,d**). The second subduction phase sees the slab entering into the lower mantle and sinking further down while progressively developing a folding pattern (**Fig. 4b–f and Fig. 5b,c,e,f**). This lower mantle subduction phase can last significantly longer because of the increased resistance to subduction due to the relatively high viscosity in the lower mantle, as well as due to the greater thickness of the lower mantle layer. Each slab fold starts to form above the 660 km discontinuity by buckling due to the resistance to downdip

slab sinking offered by the relatively high-viscosity lower mantle. Once formed the folds continue to tighten, thereby making a fold pile, which appears as a broad thermally-diffused anomaly in comparison to the well-defined slab in the upper mantle (**Fig. 5a–c**). We note that during the upper mantle subduction phase, the size of the poloidal cell that forms in the mantle wedge is relatively small, with its diameter scaling with the upper mantle thickness, whereas when the slab enters the lower mantle it activates a much larger mantle flow cell (**Fig. 4a,f**). A further point to notice is that during the lower mantle subduction phase

the slab dip angle close to the surface, in the mid-upper mantle and in the lower mantle decreases progressively with time. In





particular, the slab dip close to the surface becomes remarkably reduced after ~109 Myr of subduction to reach 7.2˚ (**Fig. 5c,f**) and continues to decrease afterwards to produce a flat slab with a 0˚ dip angle.

Changing upper mantle rheology affects both the shape of the slab at depth (**Fig. 6**) and the surficial velocities (**Fig. 7**). A Newtonian upper mantle rheology reduces the intensity of slab folding (**Fig. 6b,c,e,f**). Raising the magnitude of the non-

linearity in the upper mantle (increasing the pre-exponential factor $A$ in equation 12) increases the viscosity ratio between the lower mantle and upper mantle material surrounding the slab, thereby promoting slab folding in the upper mantle that offers less resistance to deformation. Interestingly, folding of the slab results in significant fluctuation of the subducting plate velocity ($V_{SP}$), and the amplitude of this fluctuation grows with increasing $A$, reflecting an intensification of the slab folding (**Fig. 7a–d**). In contrast, changes in trench retreat velocity ($V_T$) due to the slab folding process are much less noticeable.

Regarding surficial kinematics in relation to time, the two-phase subduction evolution is mostly noticeable in the time variation of $V_{SP}$ (**Fig. 7**). $V_{SP}$ increases rapidly during the upper mantle subduction phase to attain a peak value of which the magnitude correlates positively with the intensity of the non-linear rheology in the upper mantle. $V_{SP}$ decreases rapidly upon the slab tip reaching the 660 km discontinuity. The following lower mantle subduction phase and associated slab folding induces an oscillation of $V_{SP}$, of which the average amplitude increases significantly and the duration decreases with increasing

pre-exponential factor $A$. In addition, the average value of $V_{SP}$ is also higher for models with higher $A$. A more detailed look at the oscillation of $V_{SP}$ indicates that its amplitude varies with time although it seems to fluctuate around a plateau value during the lower mantle subduction phase for models UMlin (**Fig. 7a**) and UMnl-A1e6 (**Fig. 7b**). Moreover, during the lower mantle subduction phase the average value of $V_{SP}$ shows a progressive increase, of which the magnitude correlates positively with $A$. However, the amplitude of the $V_{SP}$ oscillation can decrease progressively with time (model UMlin, **Fig. 7a**) or it can

decrease suddenly and considerably after some time (model UMnl-A1e6, **Fig. 7b**, and reference model UMnl-ref with $A$ = 3e6 Pa$^n$ s, **Fig. 7c**). Model UMnl-A1.25e7 does not show this decrease probably because the subducting plate was entirely consumed before the stage of reduced oscillation amplitude in $V_{SP}$ was reached (**Fig. 7d**). As for $V_T$, it also rises to attain a peak value at the end of the upper mantle subduction phase. Then, during the lower mantle phase it follows different evolutions depending on upper mantle rheology. For model UMlin, $V_T$ decreases constantly until attaining slightly negative values after

~185 Myr. For models UMnl-A1e6 and UMnl-ref, $V_T$ has a constant value until ~100 Myr before it decreases to slightly above zero cm/yr on average in model UMnl-A1e6 and ~0.25 cm/yr in model UMnl-ref. On the other hand, model UMnl-1.25e7 shows a slightly increasing trend for $V_T$ during the lower mantle subduction phase and no following decrease. Furthermore, as for $V_{SP}$, the average value of $V_T$ increases with increasing $A$, meaning that the overall surficial kinematics is affected by upper mantle rheology.

The slab dip angle $\alpha_{dip}$ close to the surface (60–120 km depth), in the mid-upper mantle (300–600 km depth) and the dip angle of the slab fold pile in the lower mantle (1000–1800 km depth) also exhibit a progressive change with time, which is most evident during the lower mantle subduction phase (**Fig. 8a**). The maximum variation of $\alpha_{dip}$ is observed close to the surface, where the slab progressively unsteepens from early-stage values at 51.5–62.2˚ for all models until reaching values below 10˚ (flat slab as defined by Schellart (2020)) after ~97–113 Myr of subduction and even negative values after ~107–





124 Myr. $\alpha_{dip}$ in the mid-upper mantle also decreases and moreover oscillates over time due to the folding of the slab above the 660 km discontinuity, with the models with higher $A$ producing a greater amplitude of oscillation. During the upper mantle subduction phase, it ranges between 63.8˚ and 92˚ and it can thereafter decrease to 22˚ at the lowest (model UMnl-A1.25e7). It can furthermore be observed that the Newtonian upper mantle model leads to higher values of $\alpha_{dip}$ in the mid-upper mantle and less fluctuation with time than the non-Newtonian upper mantle models. As for $\alpha_{dip}$ in the lower mantle, it decreases quickly from 84˚ to 52˚ in model UMnl-A1.25e7. However, models UMnl-ref and UMnl-A1e6 show a much smaller and gradual decrease, with values ranging between 48˚ and 57.5˚. In addition, model UMlin produces a slightly steeper slab in the lower mantle than models UMnl-ref and UMnl-A1e6.

As to overriding plate deformation in the forearc and backarc domain, all models show the same trend of evolution with a first phase of extension and a second phase of shortening, since $V_T$ is greater than $V_{OP\_tz}$ and vice versa after some time (**Fig. 7**). During the phase of overriding plate shortening, the shortening rate increases progressively to reach a maximum after the development of a flat slab close to the surface. For the Newtonian upper mantle model (UMlin), the change from extension to shortening coincides with the start of the lower mantle subduction phase (**Fig. 7a**). However, for all non-Newtonian upper mantle models, the slab has already entered into the lower mantle before the overriding plate begins to experience its phase of shortening, and the duration of the extension phase increases with decreasing $A$. We measured the horizontal deviatoric normal stresses $\sigma_{XX}$ in the top layer of the overriding plate at an equivalent stage of subduction corresponding to a convergence of ~4000 km in all models (**Fig. 9**). For this late stage of subduction, all the models produce horizontal deviatoric compression within nearly the entire overriding plate, with a much higher magnitude in the region extending from the trench to the backarc-far backarc transition zone. The deviatoric compression reaches a maximum value ranging between ~-68 and ~-120 MPa, which is observed between ~30 and ~300 km away from the trench depending on the model. To investigate the causal relationship between this deviatoric compression and subduction-induced mantle flow, we also measured the horizontal deviatoric shear stresses $\sigma_{YX}$ in the mantle just below (5–20 km) the base of the overriding plate. For all the models, the deviatoric shear stresses show a maximum of 1.1–4.4 MPa, which corresponds in location, or is slightly offset, to the maximum deviatoric compression observed in the overriding plate. The magnitude of the deviatoric shear stresses decreases away from the trench to reach a minimum at the backarc-far backarc transition zone and increases below the cratonic domain, although with a value that remains lower than the maximum observed in the forearc-backarc domain. Remarkably, overriding plate strain rate does not seem to be greatly affected by upper mantle rheology (**Fig. 7**), except for the model with the highest $A$ (**Fig. 7d**) but this model should run longer to reach the phase of slab flattening after which the maximum shortening rate is generally observed (**Fig. 7a-c**). For the three other models with lower $A$ and a Newtonian upper mantle, increasing $A$ leads to slightly lower deviatoric compression in the overriding plate (maximum $\sigma_{XX}$ decreasing from ~-120 to ~-105 MPa) and lower deviatoric shear stresses at its base (maximum $\sigma_{YX}$ decreasing from ~4 to ~2.75 MPa) (**Fig. 9a**).

**3.2. Effects of subduction interface yield stress**



Yield stress at the subduction interface and thus subduction interface strength does affect slab shape (**Fig. 10**), slab dip angle (**Fig. 8b**) and surficial velocities (**Fig. 11**). Notably, the process of slab flattening close to the surface is greatly controlled by the subduction interface yield stress. Decreasing $\sigma_y$ indeed enhances formation of a longer flat slab, which reaches ~300, 250 and 155 km at the same amount of convergence (~4000 km) for models with $\sigma_y$ of 7, 14 and 21 MPa, respectively (**Fig. 10a-c**). Models with $\sigma_y$ of 28 MPa and 35 MPa do not produce a flat slab because they take too much time to decrease the slab dip angle close to the surface (**Fig. 10d,e** and **Fig. 8b**). Slab dip angle further deep is also affected by subduction interface yield stress but to a lesser extent. For the model with the weakest subduction interface the dip angle of the slab fold pile in the lower mantle is considerably reduced compared to the other models. In addition, for the other models ($14 \leq \sigma_y \leq 35$ MPa) the dip angle of the slab fold pile slightly decreases with increasing $\sigma_y$. As for $\alpha_{dip}$ in the mid-upper mantle, it increases with increasing $\sigma_y$ during the upper mantle subduction phase but it does not appear to be affected during the lower mantle subduction phase.

Surficial velocities are affected by $\sigma_y$ with $V_{SP}$ and $V_T$ showing a general reduction, and $V_{SP}$ showing initially a lower amplitude and higher time period of oscillation, when the yield stress is increased (**Fig. 11**). However, the model with a yield stress of 28 MPa (**Fig. 11d**) shows a progressive increase of $V_{SP}$ with time that becomes more significant than models with $\sigma_y$ of 14 MPa and 21 MPa (**Fig. 11b,c**). This leads the model with a yield stress of 28 MPa to ultimately reach fastest $V_{SP}$ after ~100 Myr, as well as a higher amplitude of oscillation in $V_{SP}$. Furthermore, the phase of initial strain rate-dependent weakening at the subduction interface at the beginning of the upper mantle subduction phase is achieved in a longer time for higher yield stresses. Similarly, the phase of progressive reduction in $V_T$, and possibly of decrease in the amplitude of $V_{SP}$ oscillation, is attained after a longer time for higher $\sigma_y$. What is more is that this phase of progressive decrease in $V_T$ is less abrupt when $\sigma_y$ is decreased.

A weaker subduction interface causes higher overriding plate extension rates during the extension phase (**Fig. 11**). As for the shortening phase, an effect is observed although it is not extreme. For models with $\sigma_y$ = 7–21 MPa, increasing subduction interface strength produces greater deviatoric compression in the forearc and backarc domain (maximum $\sigma_{XX}$ increasing from ~-76 to ~-105 MPa) and also larger deviatoric shear stresses at the base of the overriding plate (maximum $\sigma_{YX}$ increasing from ~1.9 to ~2.75 MPa) (**Fig. 9b**). However, the two models with a stronger subduction interface show a reduced overriding plate deviatoric compression and deviatoric shear stresses below the forearc and backarc compared to models with $\sigma_y$ = 14 and $\sigma_y$ = 21 MPa. Notably, the model with $\sigma_y$ = 35 MPa produces maximum deviatoric compression in the forearc, not in the backarc nor in spatial correlation with the maximum deviatoric shear stresses at the base of the overriding plate like it is observed in the other models.

### 3.3. Effects of slab thermal weakening

Rheological weakening of the slab due to progressive thermal warming also impacts on slab shape (**Fig. 12**) and slab dip angle (**Fig. 8c**) but to a lesser extent than on surficial velocities (**Fig. 13**). Increasing slab thermal weakening (increasing $E'$ in equation 14) indeed induces an intensification of the folding of the slab when it enters into the lower mantle (**Fig. 12**). Slab dip angle is, however, only slightly affected with mostly the dip angle of the slab fold pile in the lower mantle being




increased by ~8–9° at a similar stage of the model (flat slab at the surface) when the rheological weakening is activated (**Fig. 8c**). An additional effect is that the progressive reduction of the slab dip angle close to the surface is faster in models with slab thermal weakening. The flat slab geometry is reached in ~60 Myr and ~47 Myr for the models with $E' = 3$ and $E' = 13$, respectively, whereas it is attained in ~99 Myr in the reference model (**Fig. 8c**). Thus, the higher $E'$, the quicker the flat slab is attained.

Increasing the intensity of the rheological weakening due to temperature allows to considerably increase the surficial velocities, in particular $V_{SP}$ (**Fig. 13**). During the lower mantle subduction phase, $V_{SP}$ oscillates between ~1.4 and ~4.3 cm/yr in the reference model without slab thermal weakening, whereas it ranges between ~2.2 and 7.4 cm/yr and between ~3.7 and 9.5 cm/yr in the model with $E' = 3$ and in the model with $E' = 13$, respectively. However, $V_T$ is affected to a lesser extent, leading the models with higher $E'$ to have a slightly increased subduction partitioning ratio. In addition, the amplitude of

oscillation in $V_{SP}$ due to slab folding increases with increasing $E'$ while the period decreases. Furthermore, the oscillation in $V_T$ becomes negligible, thereby reducing the oscillation amplitude in the subduction partitioning ratio.

Increasing slab thermal weakening moreover increases slightly overriding plate shortening rates compared to the reference model (**Fig. 13 and 7c**) but it does not affect extension (**Fig. 9c**). Decreasing slab viscosity due to temperature leads to slightly greater deviatoric compression in the forearc and backarc domain (maximum $\sigma_{XX}$ increasing from ~-105 to ~-127

MPa) and also larger deviatoric shear stresses at the base of the overriding plate (maximum $\sigma_{YX}$ increasing from ~2.75 to ~4.4 MPa) (**Fig. 9c**).

### 3.4. Model-nature comparison

In terms of similitude with nature, the models investigating upper mantle rheology reproduce the low values of $\alpha_{dip}$ close to the surface except for model UMnl-A1.25e7 for which the subducting plate was entirely consumed before the slab dip

could be reduced further (**Fig. 8a**). For $\alpha_{dip}$ in the mid-upper mantle, part of its oscillation curve fits the natural values. As for $\alpha_{dip}$ in the lower mantle, part of its values also matches those of the natural prototype, although those occurring coeval with the phase of slab flattening close to the surface are slightly lower. The similarity regarding surficial kinematics is good if one considers the subduction partitioning ratio, where part of the curve produced by the models fits the natural range and it also shows an increasing trend (**Fig. 7**). Nevertheless, the range produced by the models is higher than in nature, as well as the

amplitude due to slab folding (**Fig. 3a**). Furthermore, increasing the pre-exponential factor $A$ increases $V_T$ and improves its fit. However, when $V_T$ decreases after the slab flattening phase it becomes much lower that the natural estimate (**Fig. 7b,c**). Most importantly, $V_{SP}$ remains much lower than its natural estimate for all models except for model UMnl-A1.25e7, but the amplitude of oscillation in $V_{SP}$ seems too large compared to the natural range in this model (**Fig. 7d**). This set of models shows that a non-Newtonian rheology in the upper mantle enhances folding of the slab above the 660 km discontinuity and it produces

a relatively more realistic oscillation of $V_{SP}$. If one looks at the evolution of $V_T$ and $V_{SP}$ in nature during the last 45 Myr, it can be seen that $V_T$ decreases progressively while $V_{SP}$ fluctuates significantly at elevated values (**Fig. 3a**). Models UMnl-A1e6 and UMnl-ref do reproduce a similar pattern after ~100 Myr of subduction evolution (**Fig. 7b,c**) except that $V_{SP}$ is too low.



In the model group investigating subduction interface strength, a low subduction interface yield stress of 7 MPa favours an increase in surface speeds so that $V_{SP}$ attains natural estimates, but the amplitude of the oscillation due to slab folding is too

great and the subduction partitioning ratio is too low (**Fig. 11a**). Moreover, this model experiences a phase of plate separation that lasts ~25 Myr at the beginning of the model (**Fig. 11a**) and the dip angle of the slab fold pile in the lower mantle is much lower than natural estimates. As for models with the highest yield stress values (28 and 35 MPa), they take a significant amount of time to reduce the slab dip angle close to the surface (**Fig. 8b**). This leaves models with a subduction interface yield stress of 14 and 21 MPa optimal candidates to reproduce the dip angle values of the slab (**Fig. 8b**) and some aspects of surficial

kinematics, including the subduction partitioning ratio, the amplitude of oscillation in $V_{SP}$ and the phase of progressive reduction in $V_T$ (**Fig. 11b,c**).

The surficial velocities obtained with the models that include slab thermal weakening provide an improved fit with the natural case, in particular for the highest $E'$ (**Fig. 13e**). Notably, from ~37 Myr of subduction evolution, $V_{SP}$ fluctuates nearly entirely inside the range of natural estimates and $V_T$ displays a progressive decrease with time comparable to nature (**Fig. 13e**

and **Fig. 3a**). On the other hand, a difference remains with the South American subduction zone as the average period of oscillation in $V_{SP}$ is shorter (~6.7 Myr) than in nature (~16 Myr on average). An improved fit is also observed on the values of dip angle of the slab fold pile in the lower mantle (**Fig. 8c**).

## 4. Discussion

### 4.1. General results

Our models show a polyphase evolution of subduction characterised by changes after the slab has started to penetrate into the lower mantle. The two-phase subduction evolution observed in our models that include a lower mantle reservoir indeed controls to a great extent the slab geometry, surficial kinematics, and overriding plate deformation, as previously shown in numerical subduction models (Faccenna et al., 2017; Schellart, 2017). Notably, slab folding develops during the lower mantle subduction phase due to the resistance to penetration into the lower mantle. This type of slab folding occurring at or above the

660 km discontinuity was also investigated in earlier numerical and analogue models, in which a relatively high viscosity ratio between two mantle layers (Griffiths et al., 1995; Guillou-Frottier et al., 1995; Ribe et al., 2007; Lee and King, 2011; Čížková and Bina, 2013; Cerpa et al., 2014; Garel et al., 2014; Schellart, 2017), or a rigid bottom (Ribe, 2003; Schellart, 2005; Guillaume et al., 2009; Gibert et al., 2012; Cerpa et al., 2014), promotes slab folding. Using tomographic images to detect slab folding/buckling at the upper-lower mantle transition and in the lower mantle in nature is not straightforward. Nevertheless,

tomography models image the South-American slab as a broader high-velocity anomaly in the lower mantle and in the mantle transition zone relative to the upper mantle (Amaru, 2007; Simmons et al., 2012; Lu et al., 2019). We propose that this broader anomaly images a slab fold pile as those produced in our numerical subduction models. As earlier modelling work (see above), our study supports the hypothesis that slab folding/buckling can occur at the 660 km discontinuity. We propose that slab buckling is a generic process that occurs elsewhere, as it was suggested at other subduction zones (Tonga, Izu-Bonin, Solomon)





based on mapping of deep earthquakes occurring in the hinge zones of the slab folds above the upper-lower mantle discontinuity (Myhill, 2013) and interpretations of seismic tomography images (Schellart et al., 2006; Ribe et al., 2007).

A second significant geometrical change relates to the progressive decrease in slab dip angle close to the surface, ultimately leading to the production of a flat slab in most models. Our models are dynamically (buoyancy) driven, thus showing that slab flattening is possible without necessarily adding other external factors such as an externally forced overriding plate
trenchward motion, as was proposed earlier (van Hunen et al., 2004; Schepers et al., 2017). Similarly, our results show that local perturbation of the subducting lithosphere buoyancy by addition of a positively buoyant aseismic ridge or plateau (Gutscher et al., 2000; van Hunen et al., 2000) is not needed to produce a flat slab. Our models instead produce a flat slab by progressive decrease of the slab dip angle in a very long time (~>36–116 Myr depending on model parameters). This result is in agreement with recent internally-driven numerical subduction models suggesting that flat slab is a feasible process at
subduction zones that are active for a very long time (~>80–110 Myr) and have a very long trench-parallel extent, greater than ~6000 km (Schellart, 2020). Our models, being two-dimensional, represent an end-member scenario of an infinitely wide subduction zone and thus fall in this wide slab category. The mechanism of flat slab development in such a geodynamic context was explained in Schellart (2020). Subduction-induced mantle flow at the scale of the whole mantle occurs during the lower mantle subduction phase. This large-scale mantle flow generates a "forced" trenchward motion of the overriding plate and it
promotes the development of high vertical deviatoric tensional stresses in the mantle wedge, which together progressively reduce the slab dip angle while the increased mantle wedge suction force helps to maintain the flat slab. Our models with a non-linear upper mantle and no slab thermal weakening further indicate that slab dip angle in the mid-upper mantle and the dip of the slab fold pile in the lower mantle can also decrease progressively with time but to a lesser extent.

As soon as the lower mantle subduction phase starts, slab folding initiating above the 660 km discontinuity generates a
strong oscillation in $V_{SP}$. This oscillation in subducting plate velocity or convergence rate due to slab folding was also observed in previous numerical subduction modelling (Lee and King, 2011; Gibert et al., 2012; Čížková and Bina, 2013; Cerpa et al., 2014; Schellart, 2017) and was suggested to occur for younger oceanic lithosphere, which is more prone to deform (Goes et al., 2008). In our models, the period of $V_{SP}$ oscillation varies between ~6 and ~21 Myr, depending on the independent variable values defining upper mantle rheology, subduction interface strength and slab thermal weakening. In earlier modelling work,
the period of velocity oscillation varies between 15 and 35 Myr depending on yield stress in the slab, lower-mantle viscosity and subducting plate crustal viscosity (Čížková and Bina, 2013), 20 and 40 Myr depending on upper-lower mantle layer viscosity jump (Lee and King, 2011; Cerpa et al., 2014), overriding plate properties (Gibert et al., 2012), and slab strength (Lee and King, 2011), and it is ~21 Myr in Schellart (2017). The period of oscillation in our models is thus comparable to previous modelling studies, although it is in the lower range or it can be slightly lower. We note, however, that it compares
well with the two to three time periods of oscillation that range between ~10–22 Myr in nature according to our calculation of the Nazca plate velocity (**Fig. 3a**).

Another notable change is the progressive decrease in trench retreat velocity occurring several million years (40–100 Myr) after the start of the lower mantle subduction phase. Such slowing down of the trench retreat as approximated by the





westward drift of the South American plate is visible in our kinematic restoration (**Fig. 3a**) and was documented in previous

numerical subduction models (Faccenna et al., 2017) and kinematic reconstructions (Sdrolias and Müller, 2006; Müller et al., 2008). In Faccenna et al. (2017) the reduction in $V_T$ occurs soon after the start of the lower mantle subduction phase. This is different from our model results, in which the decrease starts a longer time after the slab has penetrated into the lower mantle. We relate this difference to the occurrence of slab folding in our models whereas the models from Faccenna et al. (2017) produce much less slab folding. The link between the progressive reduction in trench retreat velocity and the occurrence of

slab folding is best visible in our **Figure 11**. Indeed, we can see that the progressive decrease in $V_T$ is concurrent with the reduction in amplitude of $V_{SP}$ oscillation and that the simultaneous occurrence of these two processes takes place at a later time for a stronger subduction interface (**Fig. 11b,c**) or does not occur at all (**Fig. 11d,e**). We propose that when the slab accommodates the resistance to penetration due to the high viscosity in the lower mantle by folding, the slab can continue to rollback. In contrast, when the slab folding fades out, the slab rollback rate decreases to accommodate the resistance to sinking

in the lower mantle ("anchored" slab conceptual model). Why does slab folding stop or decrease in some models may be explained as follows. When the resistance to slab penetration into the lower mantle due to the viscosity or density increase is greater than the pull force of the slab fold pile, folds form to accommodate the residual force resisting slab penetration in the direction of slab dip. After further subducting plate consumption into the mantle and growth of the slab fold pile, the pull force of the slab fold pile overcomes the resistance to slab penetration into the lower mantle and the slab does not need to deform

anymore. However, the rollback rate decreases due to the higher resistance to lateral motion because in the lowermost part of the upper mantle the slab is no longer allowed to have a lateral migration component as it is pulled into the lower mantle by the slab fold pile. We note that in our models that include slab thermal weakening the oscillation in $V_{SP}$ and associated slab folding do not vanish but decrease, and the reduction in $V_T$ occurs concurrently, similarly as in nature.

Our models furthermore agree with earlier numerical modelling research showing that overriding plate deformation

switches from extension to shortening after the slab has started to sink into the lower mantle (Faccenna et al., 2017; Schellart, 2017). Extension occurs during the upper mantle subduction phase due to slab rollback and progressive slab bending and associated increase in slab dip angle. Moreover, during this phase, a mantle flow poloidal cell forms in the mantle wedge and its size scales with the upper mantle thickness. Thus, the mantle flow-induced shear stresses responsible for dragging the overriding plate at its base (and deforming it) are maximum within 660 km away from the trench, thereby generating overriding

plate extension. In contrast, once the slab has started to sink into the lower mantle, the mantle flow occurs on a much larger dimension scaling with the whole mantle. This whole mantle flow drags the entire overriding plate towards the trench and promotes shortening. Shortening in the overriding plate backarc does not necessarily start to occur soon after the slab has started to sink into the lower mantle. It can indeed take up to several tens of Myr before it initiates, thus displaying a rather progressive change as observed in previous studies (Faccenna et al., 2017; Schellart, 2017).

**4.2. Effects on slab geometry**



Our results show that slab folding and slab dip angle are significantly affected by all our tested parameters except for slab thermal weakening that has only a moderate impact on slab folding.

In our models, the hundred-fold increase in viscosity between the upper and lower mantle leads the whole portion of the slab that sunk into the lower mantle to deform by folding. Our study indicates that upper mantle rheology and slab thermal weakening provide control on the intensity of slab folding. A power-law rheology in the upper mantle favours slab folding by decreasing the effective viscosity in regions of high strain rates, thus increasing the slab to upper mantle viscosity ratio locally around the slab. Additionally, slab thermal weakening reduces the effective viscosity of the subducting plate, thereby making it less resistant to deformation and more prone to fold. Such impact of lithospheric strength on slab deformation and folding was shown in earlier numerical modelling work that varied the viscosity of the slab (Lee and King, 2011), its temperature (Arredondo and Billen, 2016), or the age of the subducting plate, thus its strength via the age-temperature relationship (Čížková and Bina, 2013; Garel et al., 2014). Subduction interface yield stress provides additional but moderate control on slab folding, since only the model with the highest yield stress of 35 MPa noticeably reduces the intensity of slab folding, while the model with the lowest yield stress of 7 MPa mostly modifies the geometry of the slab fold pile without changing much the folds themselves. While at intermediate yield stresses the geometry of the slab folds is not affected, the time period of slab folding varies with subduction interface strength (see 4.3).

Slab dip angle close to the surface and the occurrence of slab flattening depend greatly on subduction interface yield stress, whereas upper mantle rheology and slab thermal weakening have little to no influence since they do not prevent the development of slab flattening nor its efficiency. Our models provide new insights on the effect of subduction interface strength on flat slab development. Indeed, a weak subduction interface is shown to promote slab flattening with a flat slab attained sooner and having a larger trench-perpendicular extent (**Fig. 10, 11**). We propose that a weaker subduction interface produces much faster subducting plate and subduction velocity, and thus a faster whole mantle return flow, causing faster overriding plate trenchward velocities, and thus faster trench pushback. Thereby, flat slab subduction occurs earlier and is more significant.

Regarding the deeper slab geometry, subduction interface yield stress impacts on the slab dip angle in the mid-upper mantle and slightly on the dip angle of the slab fold pile in the lower mantle, whereas upper mantle rheology and slab thermal weakening have a very minor effect. In our models, the slab dip in the mid-upper mantle is steeper for stronger subduction interfaces. This is consistent with earlier subduction modelling studies showing that higher subduction interface strengths lead to higher dip angles in the upper mantle (Čížková and Bina, 2013; Duarte et al., 2013; Holt et al., 2015). However, in the four models with higher subduction interface yield stresses, the correlation between $\sigma_y$ and dip angle of the slab fold pile in the lower mantle is slightly negative. Only the lowest $\sigma_y$ produces a much lower dip angle of the slab fold pile in the lower mantle than the higher $\sigma_y$. This result indicates that the subduction interface strength does not significantly control the dynamics of subduction in the lower mantle at reasonable yield stress values (~14–21 MPa).

### 4.3. Effects on surficial velocities



Our study indicates a significant control of all tested independent variables on surficial velocities. The mean value of
$V_{SP}$ and $V_T$ is controlled by all parameters. A power-law upper mantle leads to faster average $V_{SP}$ due to reduced effective
viscosity in regions of high strain rates, as previously shown in earlier convection and subduction models (Christensen and
Yuen, 1989; van den Berg et al., 1993; Billen and Jadamec, 2012). Moreover, our models return faster trench retreat rates
when increasing the intensity of the upper-mantle non-linearity. This is opposite to a previous 2-D and 3-D subduction
modelling study (Holt and Becker, 2017), in which a power-law mantle rheology produces slower trench retreat rates compare
to a Newtonian mantle. The authors propose the mantle shear force is the dominant factor resisting downdip slab motion and
that it is reduced to a greater extent than the pressure gradient across the slab, which is the dominant factor resisting slab
rollback. We infer that this different outcome is due to the different model setup between the two studies. The models from
Holt and Becker (2017) extend down to the 660 km discontinuity, whereas our models extend down to the core-mantle
boundary. Therefore, the sub-slab pressure in their upper-mantle models is higher than in our study, making this resisting force
a primary factor that can slow down slab rollback while downdip slab sinking is less resisted due to the reduced viscosity
around the slab with a non-linear upper mantle. Looking closely at our results (**Fig. 7**), the increase in trench retreat rates is
observed mostly during the lower mantle subduction phase, whereas during the upper mantle subduction phase, which best
compares with the models from Holt and Becker (2017), $V_T$ is similar for all tested upper mantle rheologies, making a better
agreement between the two studies. It remains to explain why during the lower mantle subduction phase $V_T$ increases with
increasing the non-linearity in the upper mantle. We propose that this is a direct consequence of the higher $V_{SP}$, which produces
a higher "forced" overriding plate trenchward velocity due to the faster upper-mantle subduction-induced mantle flow. An
additional explanation may be that in the Newtonian upper mantle model slab flattening occurs earlier. Thus, it shortens the
overriding plate and contributes to reduce $V_T$.

Furthermore, our results show that similarly as with upper mantle rheology, decreasing subduction interface strength
increases $V_{SP}$, which in turn induces faster mantle flow velocities because the viscous resistance to downdip sinking at the
interface is reduced, in agreement with earlier numerical subduction modelling (Čížková and Bina, 2013; Holt et al., 2015).
The faster mantle flow velocities cause more "forced" overriding plate trenchward motion, thus relatively higher $V_T$. During
the early stages of our models, lower subduction interface yield stresses lead to a lower subduction partitioning ratio, thus a
relatively higher $V_T$, but ultimately the models all attain a comparable subduction partitioning ratio. This corroborates previous
findings that a weaker subduction interface promotes faster $V_T$ that results in lower slab dip angles in the upper mantle (Čížková
and Bina, 2013; Holt et al., 2015). However, why the subduction partitioning ratio is affected remains elusive. Further control
from slab thermal weakening provides a similar effect as subduction interface strength by decreasing the resistance to slab
folding and downdip slab sinking due to a thermally reduced slab viscosity. This is in agreement with Lee and King (2011)
showing faster convergence rates for weaker slabs.

How the tested parameters affect the oscillation in subducting plate velocity relates directly to their control on the slab
folding process (see 4.2). Models with stronger non-linearity in the upper mantle produce a greater amplitude and a reduced
time period of oscillation. This is consistent with numerical upper-mantle subduction models from (Cerpa et al., 2014) in which



weaker Newtonian upper mantle models decrease the oscillation time period. Furthermore, our models indicate that increasing subduction interface strength decreases the oscillation amplitude and increases the oscillation time period since it reduces the

mean subducting plate velocity, which provides a first order control on slab folding (Gibert et al., 2012). The time period increase for stronger subduction interfaces was similarly observed in numerical subduction models from Čížková and Bina (2013). Our models including slab thermal weakening also agree with previous subduction modelling studies that show a dependence of folding periodicity on slab stiffness, regardless of whether slab effective viscosity is parametrically varied directly as an initial condition (Lee and King, 2011) or depending on its initial temperature (Arredondo and Billen, 2016), on

the age of the subducting plate (Čížková and Bina, 2013; Garel et al., 2014), or on progressively reduced temperature-dependent viscosity (this study). In all these earlier models and our study, decreasing slab viscosity (and thus slab stiffness) decreases the time period of slab folding due to a lower bending moment (Ribe, 2010).

### 4.4. Effects on overriding plate deformation

Our results indicate that the tested parameters provide some control on overriding plate deformation. Despite faster

subduction velocity and induced mantle flow when increasing the non-linearity in the upper mantle, lower deviatoric shear stresses are observed below the overriding plate because of the reduced viscosity in regions of high strain rates. The viscosity is reduced more significantly in the upper mantle than the velocity and strain rate are increased as a result of the power-law equation in equation 12. Lower deviatoric shear stresses below the overriding plate in turn induce lower deviatoric compression in the overriding plate. Furthermore, there is generally (in all models of this study) a spatial correlation between $\sigma_{XX}$ and $\sigma_{YX}$

in terms of their magnitude (**Fig. 9**), which indicates that mantle flow-induced deviatoric shear stresses at the base of the overriding plate are a primary cause of overriding plate deformation, as was previously suggested in analogue and numerical subduction modelling studies (e.g. Schellart and Moresi, 2013; Holt et al., 2015b; Chen et al., 2016). Our model with Newtonian upper mantle rheology further shows that the switch from extension to shortening is concurrent with the start of the lower mantle subduction phase, whereas in the models with a power-law upper mantle rheology, the switch occurs later.

This may be a consequence of a slight decoupling occurring between the two mantle layers when a non-linear rheology is used in the upper mantle.

For the models with $\sigma_y = 7$–21 MPa, despite higher mantle flow velocity for weaker subduction interface the deviatoric compressive stresses in the overriding plate are not higher. This result differs from earlier numerical subduction models (Holt et al., 2015) in which a reduced subduction interface strength using visco-plasticity produces greater overriding plate

compression than with the reference Newtonian interface. However, it is more consistent with analogue subduction models that promote backarc extension with a weaker plastic or viscous subduction interface (Duarte et al., 2013; Meyer and Schellart, 2013; Chen et al., 2016) and forearc shortening with a stronger interface (Duarte et al., 2013). We note, however, that a direct comparison between this work and laboratory subduction models is limited due to the different geometric setup. Indeed, this study uses a whole mantle depth and a 2-D geometry, which approximates an infinitely wide subduction system, whereas the

above-mentioned analogue subduction models are 3-D and simulate subduction zones of limited lateral extent that subduct



into an upper mantle reservoir, in which the toroidal component of mantle flow plays a preponderant role in driving overriding plate deformation (Chen et al., 2016). In our work, the component of overriding plate deformation caused by mantle flow is due to poloidal flow only. For models with $\sigma_y$ of 28 and 35 MPa, there is no slab flattening and the mantle flow velocities are reduced due to the strong interface, which both explain the decreased deviatoric compression in the forearc and backarc. This

result is similar to the study of (Holt et al., 2015), probably because our stronger subduction interfaces correspond more to their Newtonian model, which produces slow subduction velocities.

Increasing slab thermal weakening leads to faster subduction rate and induced mantle flow, which in turn increases the deviatoric shear stresses applied at the base of the overriding plate and generate greater forearc and backarc deviatoric compression.

**4.5. Comparison with and implications for South American subduction zone**

Here we aim to compare results of our numerical subduction models with the South American subduction zone. Our models are buoyancy-driven with no external force or velocity to drive subduction. Therefore, if the models can reproduce geometries and kinematics of the natural subduction system, they will prove robust in order to investigate geodynamic processes further and test hypotheses in future modelling studies. Moreover, our modelling is independent from Andean

geological constraints, allowing us to provide implications on the evolution of the Andes. Some aspects of the surficial velocities produced in our models are comparable with the South American subduction zone. $V_{SP}$ is higher than $V_T$ and we obtain a fairly similar subduction partitioning ratio, which is an indicator that the geometrical setup, boundary conditions and rheology of the model allow to approach the natural case quite well in terms of non-dimensional kinematics. In addition, some of the models allow to reproduce a number of slab geometries that are known or thought to occur with the Farallon-Nazca

slab, namely slab dip angle reduction leading to slab flattening close to the surface, as well as slab folding at the 660 km discontinuity.

Looking into more detail, we propose to determine a best-fitting model that successfully reproduces a number of fitting criteria on surficial kinematics, slab geometry and timing of relevant observations. To achieve this goal, we define a fitting score that is based on three main criteria, two of which include sub-criteria, as follows: (1) surficial kinematics including (1a)

subduction partitioning ratio fits natural range and trend, (1b) absolute velocities are comparable with nature, (1c) amplitude and time period of $V_{SP}$ oscillation are comparable with nature, (1d) Progressive reduction in $V_T$ is reproduced; (2) slab dip angle including (2a) length of flat slab portion must be realistic, (2b) slab dip angle close to the surface is comparable with nature, (2c) slab dip angle in the mid-upper mantle is comparable with nature, (2d) the dip angle of the slab fold pile in the lower mantle is comparable with nature; and (3) timing, namely the models should show synchrony of three elements: $V_{SP}$

oscillation, progressive reduction in $V_T$ and comparable values of slab dip angle at the three depth intervals considered in this study. A score for each criterion is given between 0 and 2, where 0 is not comparable, 1 is somewhat comparable and 2 is very comparable, which gives a maximal total score of 18. We note that we have not tested the role of overriding plate rheology,





which can strongly affect deformation rate and therefore we do not use overriding plate deformation as a means to build the model fitting score.

The best-fitting model is the model with maximal slab thermal weakening, which attains a fitting score of 17 out of 18 (**Fig. 14**). The fit is better than the reference model mostly because of the increased absolute surficial velocities that can reach the range of natural estimates, notably $V_{SP}$, due to strong reduction of the effective slab viscosity and associated shear viscous stresses resisting downdip slab sinking (**Fig. 13e**). This implies that temperature-dependent slab viscosity may be a parameter of primary importance that should be considered and investigated further in geodynamic models. However, a remaining

difference of this model with the South American subduction zone is that the average time period of oscillation in $V_{SP}$ is shorter (~6.7 Myr) than in nature (~16 Myr). One possible explanation for this is that the subducting plate in our models is too weak in its centre, which promotes rapid folding and short time periods of $V_{SP}$ oscillation. We also looked into another possible reason, namely how our scaling of thermal buoyancy affects surficial velocities by running an additional model with a higher $Ra$ of ~$6 \times 10^7$. The model does produce faster $V_{SP}$ and $V_T$ than the reference model, but the amplitude of oscillation in $V_{SP}$

becomes much larger than in nature and the time period very short (**Fig. S1**). This effect would be moreover amplified using a thermal weakening of the slab. Thus, the scaling of the buoyancy force driving subduction cannot produce an improved fit. The second best-fitting models are the two models with slightly lower slab thermal weakening attaining a score of 16 out of 18, for the same reasons as above.

    Reasonable fits are also provided by the reference model reaching a score of 14 out of 18 and the model with slightly

lower subduction interface yield stress of 14 MPa (21 MPa in the reference model) reaching 15 out of 18, whereas models with lower or higher yield stress have scores ranging between 3 and 7. This result implies that only a narrow range of subduction interface yield stresses, between ~14 and ~21 MPa, is appropriate to reproduce the dynamics of the South American subduction zone. Note that the lower and upper limits might actually be between 7 and 14 MPa, and 21 and 28 MPa, respectively. This is moreover consistent with previous work suggesting that subduction zones have a weak interface of which

the strength varies very little between different subduction systems (Duarte et al., 2015), with low estimates of the shear stresses at the subduction megathrust in nature (Magee and Zoback, 1993; Kelin Wang et al., 1995; Springer, 1999; Grevemeyer et al., 2003; Seno, 2009; Luttrell et al., 2011) and with laboratory subduction models making use of very weak lubricant to sustain subduction (Duarte et al., 2014; Chen et al., 2015; Flórez-Rodríguez et al., 2019).

    An interesting point of comparison with the South American subduction zone concerns the slab folding process and

how it may produce periodic or episodic changes in deformation and topography in the Andes. Earlier analogue and numerical modelling studies have indeed suggested that the slab folding process generates a periodic change of the slab dip angle in the upper mantle, which could ultimately generate periodic changes in overriding plate deformation (Guillaume et al., 2009; Gibert et al., 2012; Cerpa et al., 2014). In these models, shallowing of the slab in the upper mantle induces overriding plate shortening, whereas slab steepening produces overriding plate extension. Our study contrasts with these studies as it does not indicate a

significant effect of the slab folding process on overriding plate deformation. However, this may be due to the relatively high-viscosity backarc domain in our models, which might suppress such a signal. Indeed, the overriding plate strain rates produced





in our models are smaller than the natural estimates, suggesting that the overriding plate in the modes is too strong. Moreover, the opposite result may be due to the different model setups. In the above studies, the models are limited vertically to the 660 km discontinuity, which is simulated as a rigid, impenetrable bottom. This differs from our models including a lower mantle 670 layer allowing for slab sinking through the 660 km discontinuity. In our models, the periodic changes in slab dip angle occurring in the mid-upper mantle are thus less significant and occur more deeply, explaining why they might not impact on overriding plate deformation since the trench and mantle wedge suction force is not affected. In addition, Gibert et al. (2012) do not incorporate a viscous mantle. Hence, their slab to mantle viscosity ratio is infinite. Furthermore, in Gibert et al. (2012) and Cerpa et al. (2014), plate motions are externally driven with velocity boundary conditions, and Guillaume et al. (2009) 675 have a 3D set-up and use a velocity boundary condition to drive subducting plate motion and use a fixed overriding plate. On the other hand, 3-D subduction modelling suggests that periodic evolution of overriding plate deformation occurs at wide subduction zones with a relatively weak overriding plate and is most pronounced away from the subduction zone centre (Schellart, 2017). Furthermore, recent modelling indicates that slab flattening can be periodic (Schellart, 2020), which could induce pronounced variations in overriding plate stresses.

**4.6. Limitations**

Our modelling contains a number of simplifications including the use of a Cartesian geometry, free-slip top boundary condition, viscously stratified subducting plate, free plate trailing edges, and the absence of mineral phase transitions. Using a Cartesian geometry is an obvious simplification of the spherical shape of the Earth, particularly considering the large model dimension. However, earlier numerical models indicate that both a Cartesian and spherical geometry produce comparable 685 subduction dynamics, induced mantle flow and slab and trench geometry (Crameri and Tackley, 2014). Furthermore, implementation of a spherical/annulus geometry was not yet stable with the version (2.7) of the code used in this study. Although the free-slip boundary condition at the top boundary of the models simplifies the free surface that is present in nature, using a free-slip top boundary has been shown to produce similar subduction dynamics, trench migration rate and slab geometry, as with a "sticky air" layer or a true free surface (Quinquis et al., 2011). Moreover, topography is not the focus of 690 our study, making free-slip a reasonable condition on the top boundary in order to optimise computing time.

Using a viscously stratified subducting plate is another simplification in comparison with the more complex rheology of the oceanic lithosphere, but earlier work showed that even plate and mantle layers with uniform viscous stratification can reproduce realistic subduction dynamics and slab geometry (King and Hager, 1990; Stegman et al., 2006; Capitanio et al., 2007; Schellart et al., 2007; Ribe, 2010; van Hunen and Allen, 2011). In particular, numerical and laboratory subduction 695 models showed that using a slab-upper mantle viscosity ratio of ~100–700 allows to reproduce the slab geometries observed on tomographic images and the associated subduction modes (Di Giuseppe et al., 2008; Funiciello et al., 2008; Schellart, 2008; Ribe, 2010). The effective viscosity of the slab in our models, thus the slab-upper mantle viscosity ratio is ~269–644 (considering minimal and maximal values of the visco-plastic upper layer), which is within the range of previous estimates. Free trailing edges of the plates may be viewed as an additional simplification. We consider they are a reasonable





approximation of the mid-oceanic ridges, which provide minimal resistance to lateral motion of the subducting plate, as widely
assumed by the subduction modelling community (Kincaid and Olson, 1987; Funiciello et al., 2006; Ribe, 2010; Chen et al.,
2016; Holt and Becker, 2017; Guillaume et al., 2018).

Our simplified implementation of the upper-lower mantle discontinuity using a viscosity jump can be considered as an
approximation of the viscosity and density changes induced by mineral phase transitions. However, subduction models that
implemented a limited number of phase transitions promote slab stagnation at the 660 km discontinuity (Čížková and Bina,
2013; Arredondo and Billen, 2016; Yang et al., 2018; Li et al., 2019). Thus, these models do not reproduce the wider range of
slab penetration modes observed on Earth and a more complete investigation of mineral phase transitions on subduction
dynamics and subduction mode is needed, which is outside the scope of this study. Considering a more complete treatment of
the phase transitions indeed allows to reproduce both slab stagnation and penetration modes (Arredondo and Billen, 2016; Li
et al., 2019). Moreover, Arredondo and Billen (2016) show that not implementing phase transitions is preferable over using
an incomplete approximation if one wishes to reproduce a realistic interaction between slabs and the 660 km discontinuity.

Spatial resolution of the models is a further aspect that we investigated in order to know if the chosen resolution is
sufficient to reproduce similar subduction dynamics as at higher resolution. We conducted two tests at higher spatial resolution
of 1450 × 725 and 1932 × 966 (number of horizontal × vertical elements) with no mesh refinement, which corresponds to 8 ×
4 km and ~6 × 3 km per element, respectively. We also ran models at lower spatial resolution. Results from this test show that
surficial velocities produced using the resolution of the present study (1024 × 512 with mesh refinement around the subduction
interface) are very close to the ones produced by the two higher-resolution models (**Fig. S2**). In our reference model, $V_{SP}$ and
$V_T$ averaged over the whole model duration are indeed a mere 0.39% ($V_{SP}$) and 0.11% ($V_T$) lower than in the highest resolution
model, which implies that our chosen spatial resolution is sufficient. Moreover, the two high-resolution models show very
comparable surficial velocities, whereas at lower resolution than the reference model the surficial velocities decrease with
reducing resolution. Thus, the two high-resolution models produce a velocity field that converges towards a unique result,
meaning that they can be used as a reference to find a lower resolution that would produce a similar result.

For this study, additional limitations concern the accuracy of our tectonic reconstruction and of the chosen tomography
models on which we base our comparison with the numerical subduction models. We used the tectonic reconstruction model
of Matthews et al. (2016) in an Indo-Atlantic hotspot reference frame in order to calculate the velocity of the Nazca plate and
of the South American plate, which is a proxy for the trench retreat rate neglecting the effect of overriding plate deformation.
Our calculated Nazca plate and South American plate velocities show comparable trends and magnitudes as in earlier tectonic
reconstruction studies that use different reference frames. For example, the Nazca plate velocity is shown to oscillate between
~2.5 and ~12 cm/yr in Sdrolias and Müller (2006), which is relatively comparable to our calculation, in which the velocity
fluctuates between ~3 and ~10 cm/yr. Similarly, the South American plate velocity gently decreases from ~2.8 to ~1 cm/yr in
the past 48 Myr in Faccenna et al. (2017), which used the tectonic reconstruction model of Seton et al. (2012). This progressive
decrease is very similar to the one calculated in this study, which occurs from ~2.5 cm/yr to ~1.4 cm/yr in the past 48 Myr.



Thus, we believe that the surficial velocities calculated in our study are robust and can represent a rigorous source of comparison for the velocities produced in our models.

Accuracy of the tomography models can be discussed in light of earlier research that uses vote maps to investigate the consistency of imaged slabs in the lower mantle (Shephard et al., 2017). Using a total of 7 P-wave and 7 S-wave models, the vote maps imply that the Farallon-Nazca slab at the centre of the subduction zone is a rather continuous feature extending down to great depths (maybe the core-mantle boundary), despite some individual tomography models like the ones chosen for this study showing discontinuities at different locations in the lower mantle (**Fig. 2e,f**). Some tomography models indeed

suggest in agreement a large discontinuity in the lower mantle and distinguish two slab-like anomalies (Li et al., 2008; Simmons et al., 2012), implying a complicated early tectonic history (Chen et al., 2019). However, the tomographic vote maps are in favour of a long-lived, continuous subduction history since the Jurassic/Early Cretaceous. This hypothesis is consistent with our and earlier numerical subduction models (Schellart, 2017) showing that a number of dependent variables can be reproduced considering an uninterrupted and long-lived subduction process, but that shows periodic velocities.

**Conclusions**

We performed a parametric study on buoyancy-driven, time-evolving, thermo-mechanical subduction models of the South American subduction zone in order to investigate progressively changing slab geometry, surficial kinematics and overriding plate deformation. We compared the first two dependent model variables with their natural counterparts using earthquake hypocentre locations, tomography studies and a tectonic reconstruction. Using a model fitting score, our approach

provides a means to determine best-fitting independent model variables for further hypothesis testing in future fully-dynamic subduction modelling. With this comparison between our model outcomes and present-day and past natural parameters we indeed show that it is possible to find a best-fitting model that therefore best approximates the dynamics of the real subduction system. Our parametric study shows that:

1. A non-Newtonian rheology in the upper mantle promotes slab folding and realistic associated oscillation of the subducting

Farallon-Nazca plate velocity.

2. The yield stress at the subduction interface must be ~14–21 MPa. A weak subduction interface is a requirement, as previously suggested by laboratory subduction models.

3. Thermal weakening of the slab must be considered if one wishes to reproduce the absolute mean surface velocities, in particular the subducting plate velocity. Such slab thermal weakening therefore deserves to be considered in future subduction

modelling.

4. Slab folding at the 660 km discontinuity can be responsible for the oscillation of the Farallon-Nazca subducting plate velocity observed in kinematic reconstructions.

5. This slab folding process does not necessarily induce periodic variations in overriding plate deformation if the overriding plate is relatively strong, as in the present study.





6. The tested parameters can affect the magnitude of overriding plate deformation, but not the general trend of switching from extension to shortening that results from switching from upper-mantle to whole-mantle subduction-induced mantle flow.

**Data availability**

The data produced for this research are available in this in-text data citation reference: Strak and Schellart (2020) under the CC BY 4.0 licence.

**Author contribution**

V.S. and W.P.S. defined the parametric study. V.S. used the code *Underworld2* to run the models and prepared a first draft of the manuscript. Both authors contributed towards analysing and interpreting the experiments and towards editing the paper.

**Competing interests**

The authors declare that they have no conflict of interest.

**Acknowledgments**

We would like to thank Arijit Laik, João Duarte, Filipe Rosas, Anouk Beniest, and Kai Xue for discussions on subduction dynamics and numerical subduction modelling. We also thank Julian Giordani, John Mansour and Arijit Laik for their technical assistance with the *Underworld2* code. This work has been funded by a Vici Fellowship (016.VICI.170.110) from the Dutch National Science Foundation (NWO) awarded to WPS. Furthermore, this work has been sponsored by NWO Exact and Natural
Sciences for the use of supercomputer facilities. The work has also been supported by computational resources from the NCI National Facility in Australia through the National Computational Merit Allocation Scheme (project qk0).

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



## Tables

| Parameter | Symbol | Dimensional value | Unit | Dimensionless value |
|---|---|---|---|---|
| Model height | $h$ | 2900 | km | 1 |
| Model length | $l$ | 11600 | km | 4 |
| Reference density (UM) | $\rho_{ref}$ | 3230 | kg m$^{-3}$ | 0 |
| SP-UM density contrast | $\Delta\rho$ | 62.985 | kg m$^{-3}$ | 1 |
| Gravitational acceleration | $g$ | 9.81 | m s$^{-2}$ | - |
| Reference temperature (UM) | $T_{ref}$ | 1300 | °C | 1 |
| SP-WM temperature contrast | $\Delta T$ | 1300 | °C | 1 |
| Thermal expansivity | $\alpha$ | 1 10$^{-5}$ | K$^{-1}$ | - |
| Thermal diffusivity | $\kappa$ | 1 10$^{-6}$ | m$^2$ s$^{-1}$ | - |
| Reference viscosity (UM) | $\eta_{ref}$ | 3.5 10$^{20}$ | Pa s | 1 |
| Lower viscosity cut-off (UM) | $\eta_{UM-min}$ | 3.5 10$^{19}$ | Pa s | 0.1 |
| Upper viscosity cut-off (UM) | $\eta_{UM-max}$ | 3.5 10$^{20}$ | Pa s | 1 |
| Lower mantle viscosity | $\eta_{LM}$ | 3.5 10$^{22}$ | Pa s | 100 |
| SP top layer viscosity | $\eta_{SP\_top}$ | 3.5 10$^{23}$ | Pa s | 1000 |
| OP crust layer viscosity | $\eta_{OP\_crust}$ | 3.5 10$^{23}$ | Pa s | 1000 |
| SP core layer viscosity | $\eta_{SP\_core}$ | 3.5 10$^{23}$ | Pa s | 1000 |
| SP bottom layer viscosity | $\eta_{SP\_bot}$ | 1.75 10$^{22}$ | Pa s | 50 |
| SP eclogitised top layer viscosity | $\eta_{SP\_eclo}$ | 1.75 10$^{22}$ | Pa s | 50 |
| OP lithospheric mantle viscosity in forearc and backarc | $\eta_{OP\_ml-FA+BA}$ | 1.4 10$^{23}$ | Pa s | 400 |
| OP lithospheric mantle viscosity in far backarc | $\eta_{OP\_ml-farBA}$ | 7 10$^{23}$ | Pa s | 2000 |
| Yield stress of SP top layer | $\sigma_y$ | 21* | MPa | 4.81 10$^5$* |
| Pre-exponential factor | $A$ | 3 10$^6$* | Pa$^n$ s | - |
| Activation energy in upper mantle | $E$ | 530 10$^3$ | J mol$^{-1}$ | - |
| Gas constant | $R$ | 8.3145 | J mol$^{-1}$ K$^{-1}$ | - |
| Activation energy in slab | $E'$ | - |  | 0* |
| Rayleigh number | $Ra$ | - | - | 4.3 10$^7$ |

**Table 1: Model parameters in reference model UMnl. An asterisk symbol indicates that tests with alternative values were conducted (see Table 2). SP, OP, WM, UM, LM, ml, FA and BA stand for subducting plate, overriding plate, whole mantle, upper mantle, lower mantle, mantle lithosphere, forearc and backarc, respectively.**






| Model name | Rheology UM | $h$ (km) | $\sigma_y$ (MPa) | $E'$ (dless) |
|---|---|---|---|---|
| UMlin | Newtonian | | | |
| UMnl-A1e6 | $A = 1\ 10^6$ | | | |
| UMnl-A1.25e7 | $A = 1.25\ 10^7$ | | | |
| **UMnl-ref** | $A = 3\ 10^6$ | **2900** | **21** | **0** |
| UMnl-yield7 | | | 7 | |
| UMnl-yield14 | | | 14 | |
| UMnl-yield28 | | | 28 | |
| UMnl-yield35 | | | 35 | |
| UMnl-E'3 | | | | 3 |
| UMnl-E'5 | | | | 5 |
| UMnl-E'7 | | | | 7 |
| UMnl-E'9 | | | | 9 |
| UMnl-E'11 | | | | 11 |
| UMnl-E'13 | | | | 13 |

**Table 2: List of tested model parameters and their values. Model UMnl-ref indicates the reference model. Nothing is indicated when the parameter is identical as in the reference model. $A$ is the pre-exponential factor in equation 12. $E'$ is the dimensionless activation energy coefficient controlling the magnitude of slab thermal weakening (see equation 14). UM indicates upper mantle. lin stands for linear, nl for non-linear. dless indicates dimensionless values.**


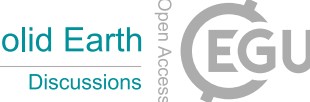

**Figures**



Figure 1: Numerical subduction model setup used in this study. (a) Compositional layering indicated using a colour coding on the Lagrangian particles. (b) Temperature and (c) viscosity stratification of the models. See methods section for a complete description of the models.





**Figure 2: Tectonic setting of the South American subduction zone and geometry of the Farallon-Nazca slab imaged from three independent tomography studies (Amaru, 2007; Lu et al., 2019; Simmons et al., 2012) and one earthquake hypocentre location study (Hayes et al., 2018). (a) Map of the South American subduction zone including a superposition of bathymetry, slab dip angle obtained from the Slab2 model (Hayes et al., 2018), and seismic velocity anomaly at 1600 km depth using tomography model TX2019slab-S (Lu et al., 2019). (b) Depth and (c) dip angle profiles of the slab using the Slab2 model and following the three profiles A–A', B–B' and C–C' reported on the map in (a). (d) Slab dip angle versus depth for the same three profiles. (e-g) Cross-sections of tomography models LLNL_G3Dv3 (Simmons et al., 2012), UU-P07 (Amaru, 2007) and TX2019slab-S (Lu et al., 2019) along profile B–B'. The red bar in (d) is an estimate of the dip angle of the positive seismic anomaly measured between 1000 and 1800 km on (e-g).**



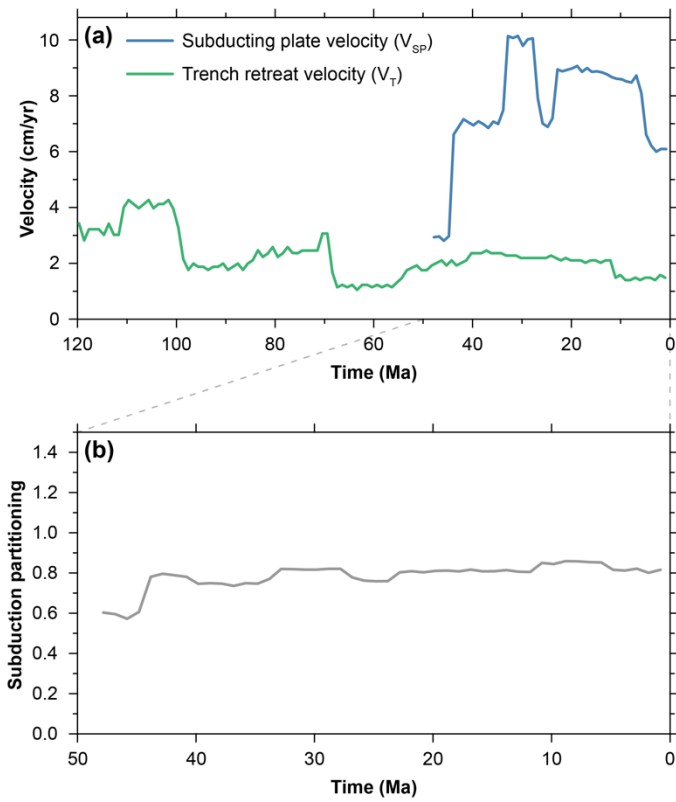

**Figure 3: Surficial kinematics of the South American subduction zone calculated using the tectonic reconstruction model of Matthews et al. (2016) in an Indo-Atlantic hotspot reference frame. (a) Farallon-Nazca subducting plate velocity and trench retreat velocity using the South American plate westward drift as a proxy. (b) Subduction partitioning ratio calculated for the past 48 Myr. The ratio is equal to $V_{SP}/(V_{SP}+V_T)$ and thus represents the proportion of convergence accommodated by subducting plate trenchward motion.**




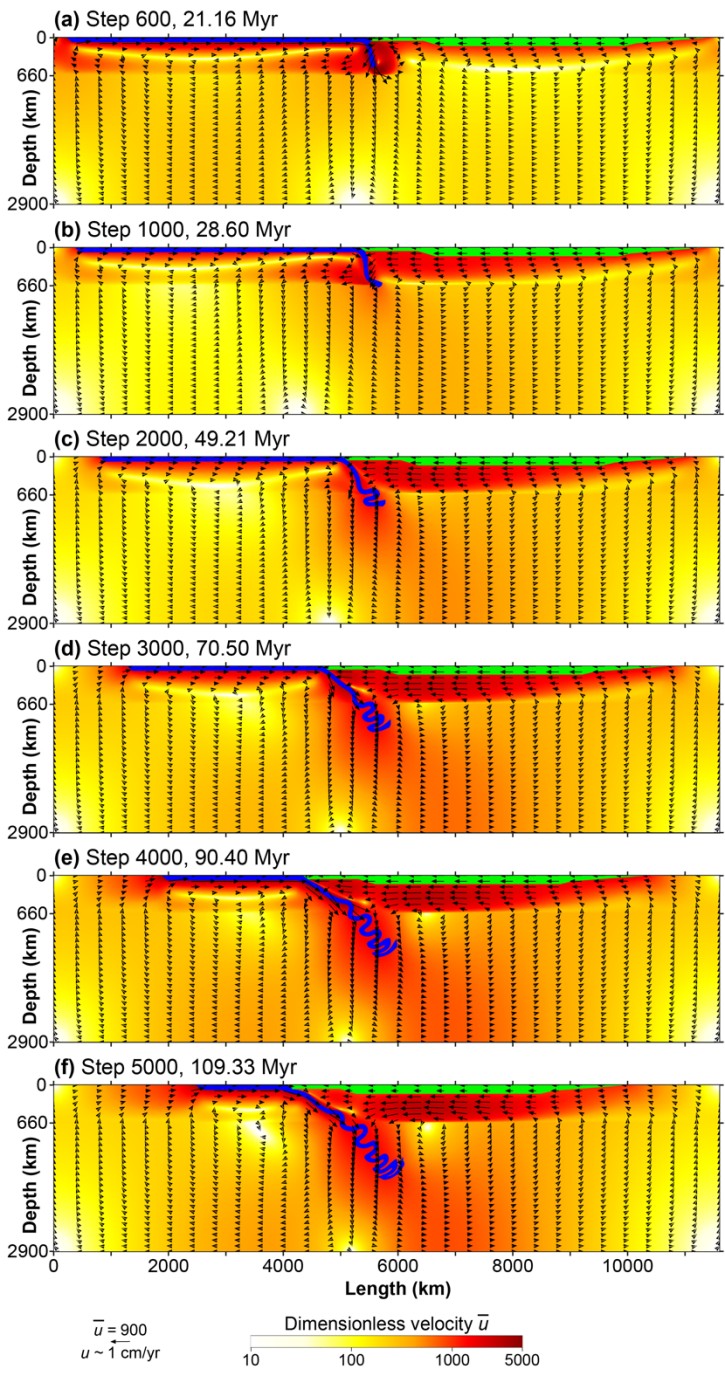

**Figure 4: (a-f) Different stages of evolution of the reference model (UMnl) showing the subduction-induced mantle flow velocity magnitude (colour scale) and orientation (vectors). The Farallon-Nazca subducting plate and slab and the South American overriding plate are indicated in blue and green, respectively.**







**Figure 5: Three different stages of the reference model (UMnl) showing the evolution of temperature (a-c) and effective viscosity (d-f).**



**Figure 6: Three different stages showing the evolution of effective viscosity in (a-c) the model with a Newtonian upper mantle (UMlin) and in (d-f) model UMnl-A1.25e7 with a power-law rheology in the upper mantle (with $A = 1.25 \times 10^7$).**



**Figure 7: Surficial velocities of models investigating the effect of upper mantle rheology. (a) Model UMlin with a Newtonian upper mantle. (b) Model UMnl-A1e6, (c) model UMnl ($A = 3\ 10^6$), and (d) model UMnl-A1.25e7 with a power-law rheology in the upper mantle. For each model we plot on the main graph the subducting plate velocity $V_{SP}$ in blue, the trench retreat velocity $V_T$ in green, the velocity at the trailing edge of the overriding plate $V_{OP}$ in dark grey, the velocity of the transition zone $V_{OP\text{-}tz}$ in light grey, and the overriding plate FA+BA deformation rate in red (extension is positive). In addition, we plot on the right end corner**
**of each main graph the subduction partitioning ratio, defined as $V_{SP}/(V_{SP}+V_T)$. The blue, green and yellow rectangles represent natural estimates for $V_{SP}$, $V_T$ and the subduction partitioning ratio, respectively. Orange triangles along the X-axis of the main graphs indicate the start of the lower mantle subduction phase (left triangle) and the initiation of slab flattening (right triangle).**



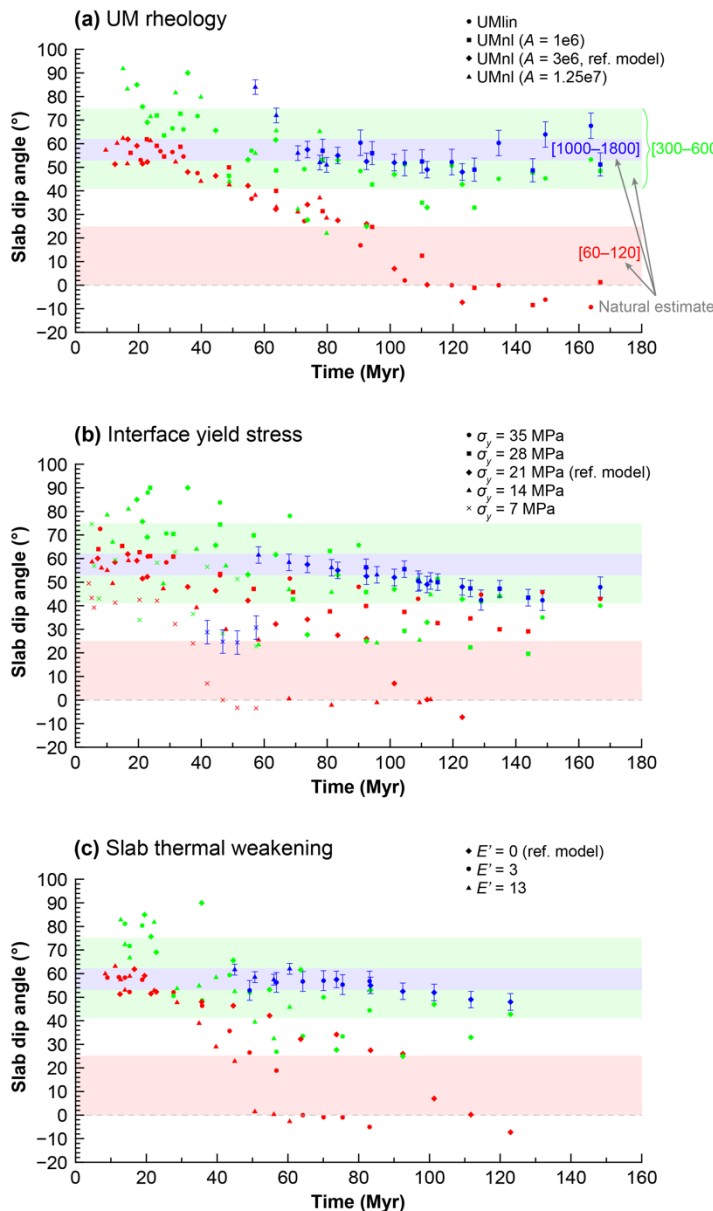

**Figure 8: Slab dip angle measured at 60–120 (red) and 300–600 (green) km depth and dip angle of the slab fold pile in the lower mantle measured at 1000–1800 km depth (blue) for all models investigating the effect of (a) upper mantle rheology, (b) subduction interface yield stress, and (c) slab thermal weakening. The coloured rectangles indicate the range of relevant natural estimates.**





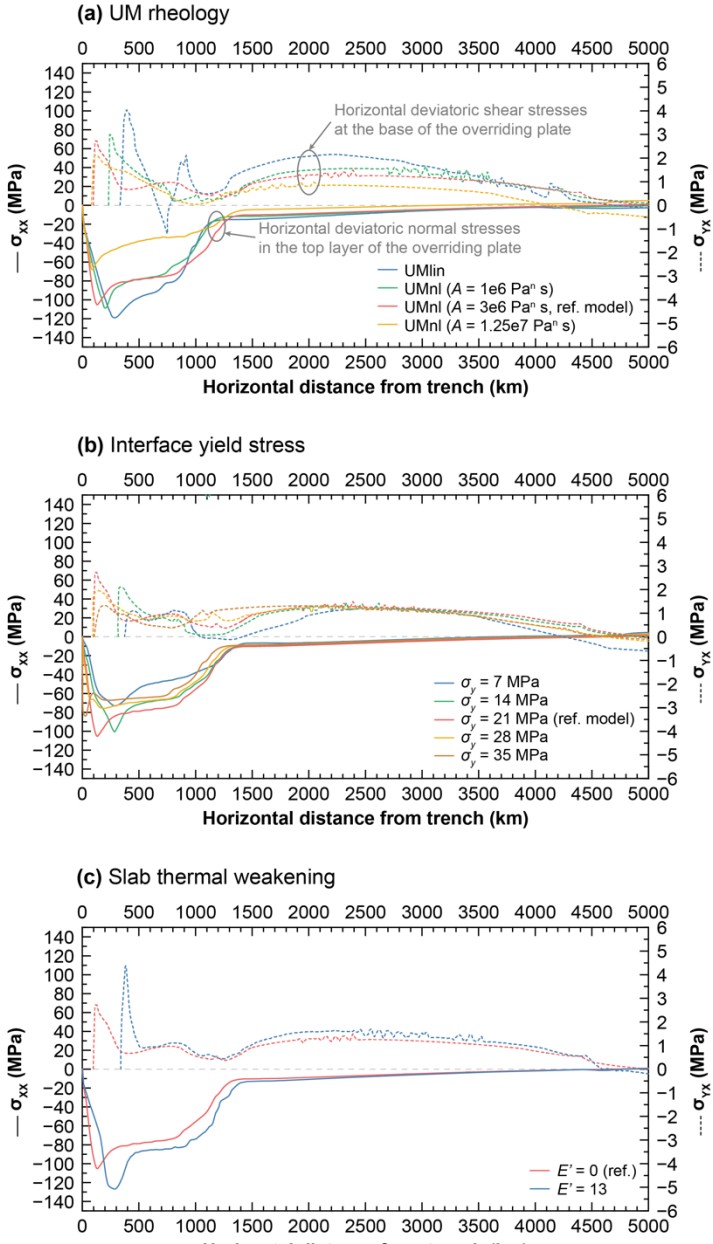

**Figure 9: Profiles of horizontal deviatoric normal stress in the top layer of the overriding plate and horizontal deviatoric shear stresses in the upper mantle just below (5–20 km) the base of the overriding plate. The profiles show the effect of (a) upper mantle rheology, (b) subduction interface yield stress and (c) slab thermal weakening. The profiles are taken at an advance stage of subduction corresponding to a convergence of ~4000 km.**




**Figure 10: Effective viscosity at an equivalent amount of convergence (~4500 km) for models investigating the effect of subduction interface yield stress set to (a) 7 MPa, (b) 14 MPa, (c) 21 MPa, (d) 28 MPa and (e) 35 MPa.**






**Figure 11: Surficial kinematics for models investigating the effect of subduction interface yield stress set to (a) 7 MPa, (b) 14 MPa, (c) 21 MPa, (d) 28 MPa and (e) 35 MPa. For more detail the reader is referred to the caption of Fig. 7.**






**Figure 12: Effective viscosity at an equivalent amount of convergence (~4500 km) for models investigating the effect of slab thermal weakening with *E'* (equation 14) set to (a) 3, (b) 5, (c) 7, (d) 9, (e) 11 and (f) 13.**



**Figure 13: Surficial kinematics for models investigating the effect of slab thermal weakening with *E'* (equation 14) set to (a) 3, (b) 5, (c) 7, (d) 9, (e) 11 and (f) 13. For more detail the reader is referred to the caption of Fig. 7.**



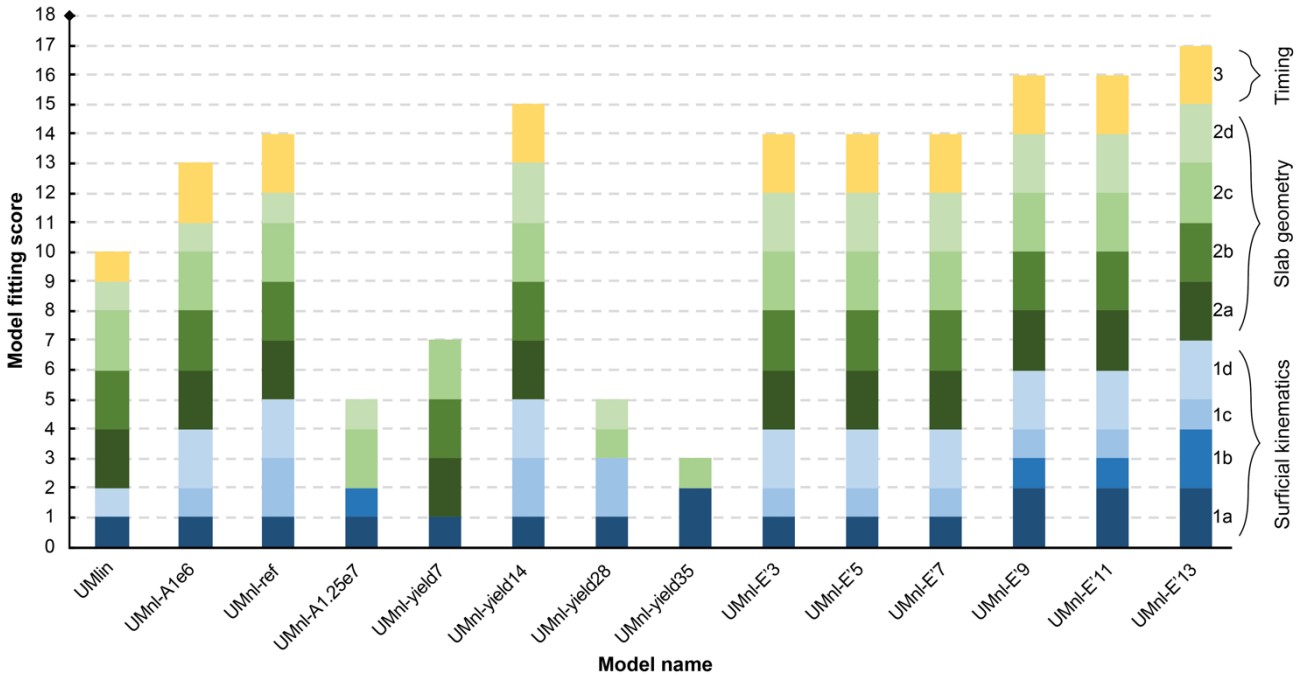

**Figure 14: Fitting score of the models presented in this study. The score is based on three main criteria, two of which include sub-criteria, as follows: (1) surficial kinematics including (1a) subduction partitioning ratio fits natural range, (1b) absolute velocities are comparable with nature, (1c) amplitude and time period of $V_{SP}$ oscillation are comparable with nature, (1d) Progressive reduction in $V_T$ is reproduced; (2) slab dip angle including (2a) length of flat slab portion must be realistic, (2b) slab dip angle close to the surface is comparable with nature, (2c) slab dip angle in the mid-upper mantle is comparable with nature, (2d) the dip angle of the slab fold pile in the lower mantle is comparable with nature; and (3) synchrony of three elements: $V_{SP}$ oscillation, progressive reduction in $V_T$ and comparable values of slab dip angle at the three depth intervals considered in this study. A score for each criterion is given between 0 and 2, where 0 is not comparable, 1 is somewhat comparable and 2 is very comparable.**