# Peer review of "Thermo-mechanical numerical modelling of the South American subduction zone: a multi-parametric investigation"

_Solid Earth, 2020_

## Referee Comment (RC1) · Anonymous Referee #1 · 15 Sep 2020

The work by Strak and Schellart uses numerical models to study the South America subduction zone. They compare their models results with data from South America using a ranking system to find which values of the studied parameters work best to fit the natural case. The manuscript is well written and the figures are clear, however I have troubles understanding the logic behind the study because I believe there is an important flaw at the base of it (see first comment below).

Major points In this study, the South America subduction zone is treated as if its characteristics (those used to rank the models) are the same everywhere along trench. This

[Figure]

is a huge assumption that I believe really affects the conclusions. Fig. 3 for example shows changes in trench and plate velocity with single lines, are these averages along the whole length of the trench, or are these taken along a specific section? If the latter is true, then which section is it? This is extremely important because, as the authors show in Fig. 2, there are large variations along trench. For instance, the slab is not flat everywhere, but there are only 2 regions where this is the case. Importantly, along section BB' the slab is not flat and this is where the positive anomaly in the lower mantle is higher (meaning that the slab pile is more clear, following the reasoning of the authors). In fact, the other two sections, where the slab is flat, do not show the same large anomaly in the lower mantle. How does this reconcile with the main conclusions and, more generally, with the philosophy of the study? In other words, this study tries to find the best 2D model that fits as many criteria as possible with the natural case, but in South America the criteria themselves are not all present in one single 2D section of the subduction zone. So, what is the point of finding a model that fits everything, when even in nature this is not the case? I used the slab dip and the slab pile in the lower mantle as examples, but also the other criteria (like the subducting plate velocity and the trench velocity) are not the same all along the trench.

Another important point is that the conclusions of this study are based on the comparison between models and data according to 9 criteria. However, at the moment, these criteria are mostly qualitative. How is the progressive reduction of trench velocity computed? What is the 'acceptable' range/error of flat slab portion length to have a rank of 1 or 2? The same goes for all criteria: what the authors define 'somewhat comparable' and 'very comparable' is subjective. I suggest to add a table in which the values of these criteria are clearly stated both for the models and for the data the models are compared with. And then again, some models might better fit one section of the subduction zone, but others that do poorly in that section might be better at fitting another section. I am not sure what we can learn from this though.

About the flat slab. The authors state that there is no need to add external forces or

have a buoyant body to have flat slab and that slab flatting can happen dynamically as a consequence of a progressive decrease of the slab dip (lines 438-443). Then one might wonder why are there only two portions of flat slab along the South America subduction zone. Why is the slab not flat along section BB' (Fig. 2)? Moreover, these models do not take into the presence of buoyant bodies in the subducting plate (which it cannot be denied it is the case in the Nazca plate with the ridges entering the subduction zone), so the question is how would the results and conclusions change if a more realistic subducting plate would be modelled?

40-44: I find the objectives of the paper very vague, they could fit with any paper that presents a parametric study. I suggest to be more specific about which features of the South America subduction zone the authors are trying to explain/understand with this study. For example, in the first sentence of the introduction it is stated that this subduction zone is enigmatic because of the orogeny formed with an oceanic subduction, but this study is not really addressing this issue. Instead, paragraph 2.5 is describing the features that are then compared with the results. To me, it makes more sense to move this paragraph in the introduction, but the authors should also add an explanation on what it is about these specific features that is not yet understood and what are the main research questions related to these features they are trying to answer.

243: from Fig. 3a it seems more that the gentle decrease in vT (between 2.5 and 2 cm/yr) is only until about 12 Ma, then there is a clear step to 1.5 cm/yr and the trench velocity remains more or less constant. Given that this is one of the thing that decides the final ranking of the models, how does this affect the results? Again, having a table with quantitative values for each criterion would help.

The amplitude of oscillation in vSP is also something that the authors look at in the models, however this is not described in paragraph 2.5 at all, but it starts to be mentioned only in the results and discussion. Describe the natural range.

The viscosity jump between upper and lower mantle has a major control on the slab

bending, piling and flattening. And slab piling and flattening is the main focus of this study. However the viscosity jump is not a parameter that is investigated. 100 is a commonly used value, but it is also an end-member. Often, other numerical studies use 30 as a viscosity jump. How do the authors think a lower viscosity jump would affect their results? I would like to see a model with a lower viscosity jump and see the effect on slab piling and flattening.

Other minor points 128: Does the assumption of a neutrally buoyant OP affect the overriding plate deformation?

The overriding plate is 60 km thick for about 1000 km from trench. How does this compare to South America?

196-198: is it only the viscosity of the slab that is affected by the thermal weakening? Or is it also the viscosity of the mantle? One of the reasons for this question is also because in Fig. 12 there are 'blue', thus very weak, regions around the slab in the lower mantle. It seems to be a consequence of numerical instability, is it?

291-292: the subducting plate does not seem to be entirely consumed in Fig. 6f, why does the model stop?

Fig. 6 only show two of the models with different A, I suggest to show the dynamics of the other two models (or at least a final snapshot) in the supplementary material.

374: "the higher E', the quicker the flat slab is attained". Or is it simply because subduction is faster, but the amount of convergence is the same?

729: then why is the natural range in figures (blue area) only going between 6 and 10 cm/yr?

---

## Referee Comment (RC2) · Anonymous Referee #2 · 16 Sep 2020

The authors present a parametric modeling study that aims to determine the physical subduction parameters associated with South American subduction. Overall, the figures are text are clear, the descriptions of the model behaviors are very well done, and the topic is interesting. However, I think there are some major issues with the study relating to i), the thermal weakening rheology adopted in a set of models, and ii), the overall study design. For the study to achieve its goals, I therefore think significant work is needed to re-think important aspects of the paper.

Specific comments

[Figure]

My first concern relates to the exploration of a thermal weakening rheology. This is important as these are the most successful suite of models. Ultimately, I am not sure your rheological implementation corresponds to what you intend. In the rheology section, you first describe a standard T-dependent Arhenius flow law (Eq. 12) and mention that the lithospheric layers have variable viscosities controlled by composition (lines ∼185). In a compositional model, you would typically neglect the temperature dependence of the viscosity in the lithosphere (with the view that the compositional strengthening is mimicking this). Because you have both lithospheric composition and cold temperature, I am not sure how you dealt with this? (But I presume you did sufficiently as the viscosity field does look reasonable in the first non-thermal weakening figures). More importantly, "thermal weakening" is then introduced as another T-dependent viscosity (Eq. 14). This expression is just a linearized version of Eq. 12 and so I am not sure: why it is referred to as thermal weakening, how it is combined with the other two viscosities (compositional and Arrhenius), and what process it describes. Looking at the figures, it has the non-intuitive effect of lowering viscosities in the cold temperature lithosphere, particularly in the lower mantle (which doesn't agree with Eq. 14). What does this correspond to physically? (The studies that you cite – Ratcliff and Schubert (1996) and Zhong et al. (2000) - just use this flow law to approximate a regular temperature dependent viscosity which, in disagreement with your models, produces strengthening in cold regions).

My other concern relates to the design of the study. The goal is the determination of geodynamic parameters for future 3-D modeling from the current 2-D models. However, the parameters chosen for exploration are not well justified. Why do you focus on these three parameters? If you are just trying to nail down a geodynamic reference setup, there are many other parameters that could significantly influence subduction and are just as uncertain: e.g. slab strength and rheology, lower mantle strength, oceanic plate density (e.g. plateaus), upper plate rheology. I think you need to either explore a larger range of parameters or provide more robust justification for your choices. Second, trying to find a 2-D reference model for a very 3-D subduction system that exhibits strong

along-strike variation (e.g. flat slab vs. no flat slab) seems challenging. I acknowledge you need a starting point for future 3-D models, and so it's probably worthwhile, but I think this produces extra concerns that should be addressed. For instance, one of the fit criteria is flat slab subduction. What about the regions that don't have flat slab subduction? Is a SAmerica reference model that produces flat subduction appropriate? (Especially given proposed links to buoyant oceanic plateaus in this region.) Also, at what latitude are the plate velocities extracted (Fig. 3) for comparison with models? Are they representative of the whole margin? For dips, you consider the along-strike range which seems very sensible. Perhaps a similar approach is also needed for the plate velocities?

Other comments

75-84: Relates to my first main comment but many studies consider temperature dependent slab viscosities (e.g. Garel et al., 2014, G-cubed) but, importantly, not in combination with a compositional slab viscosity.

100-105: If you are quoting a run times then you should also state the size of the model (e.g. total number of elements, degrees of freedom).

Eq. 5: Is compositional buoyancy just applied to the upper plate and not the subducting plate? I could not figure this out (also Lines 165-170).

~240-250: You ignore the OP shortening component of vt. Fair enough, but could this be why you get a progressive decrease in vt (Fig. 3b)? If so, is it appropriate to match this vt trend with models that don't have significant OP deformation?

503: Factor 100 lower mantle viscosity increase is probably reasonable, but on the high end. Did you test a reduced value?

532: According to what reasoning are these crustal yield stress values reasonable? Refs?

644: Ra # increased by reducing mantle viscosity? Or increasing slab/plate density?

673: What is this Gilbert paper? Not in reference list. If similar to the Cerpa work, they do not solve for the viscous mantle but parameterize it using edge forces on the slab. So not really fair to call these infinite slab-mantle viscosity ratio models.

Overall, the discussion is long winded and, in places, repetitive. Further effort to consolidate it around the main points would really improve readability!

―――――――――――――――――――

---

## Author Comment (AC1) · 22 Sep 2020

Comment The work by Strak and Schellart uses numerical models to study the South America subduction zone. They compare their models results with data from South America using a ranking system to find which values of the studied parameters work best to fit the natural case. The manuscript is well written and the figures are clear, however I have troubles understanding the logic behind the study because I believe there is an important flaw at the base of it (see first comment below).

Response We thank Anonymous Referee #1 for his review work on our manuscript and for providing a critical analysis. The main criticism raised by Anonymous Referee #1 (and also raised by Anonymous Referee #2) relates to the fact that we use a two-dimensional model setup to study a three-dimensional subduction system of which the investigated parameters vary along the trench, thus in the third dimension that is not included in our models. As discussed in our detailed responses below, this is not a major issue since the two-dimensional approach is, as a matter of fact, appropriate to study the dynamics at the centre of wide subduction zones such as for South America using a vertical section at the centre of the subduction zone. So we are really trying to model only the centre of the South American subduction zone (Bolivian orocline region), and only compare our model results with this central segment in nature, and not the segments to the north and south. This point was mentioned in the methods section of the original manuscript (L98-101). In our revised manuscript we will state this more clearly by adding/revising text in the abstract, introduction, methods and captions of Fig. 2 and 3 (see detailed responses below).

Comment Major points In this study, the South America subduction zone is treated as if its characteristics (those used to rank the models) are the same everywhere along trench. This is a huge assumption that I believe really affects the conclusions.

Response The reviewer comments that our study assumes that a number of South American subduction zone characteristics (e.g. subducting plate velocity, trench velocity, subduction partitioning, slab dip) are constant along the trench. This is not the case. We think this comment stems from an oversight by the reviewer that, with our models and our model-nature comparison, we only focus on the central segment of the South American subduction zone (lines 98-101 in the Methods of our original manuscript). Indeed, our 2D model approach dictates that it is only (approximately) applicable to the central segment of a wide (and symmetrical) subduction zone [e.g. Schellart et al., 2007; Schellart, 2020], of which the South American subduction zone is the best present-day example of the Earth. In any case, we realize now that we could have

stated this more explicitly in our manuscript. Thus, to avoid any potential confusion in the future, we now state more explicitly that our model-nature comparison is only applicable to the central segment (Bolivian orocline region) of the South American subduction zone.

To accommodate the comment from the reviewer we will revise the first sentence of the Methods section such that our revised manuscript reads:

"The regional models were designed to conduct a parametric investigation on the effect of upper mantle rheology (linearly or non-linearly viscous), subduction interface yield stress $\sigma y$ and slab thermal weakening on the subduction dynamics of the central segment (Bolivian orocline region) of the South American subduction zone over a long timescale ($\sim$60-200 Myr) and large spatial dimensions (11600 km laterally and 2900 km vertically)."

We have also modified the sentence on lines 98-101 in the original manuscript by adding several references that support our claim:

"The 2-D approach is a reasonable approximation considering that we simulate the subduction process at the centre of a very wide subduction system where toroidal mantle flow is minimal [Schellart et al., 2007; Schellart, 2017] and slab geometry and plate kinematics are very similar as in 3-D subduction models at the centre of the subduction system [Schellart, 2020]."

We have also added another sentence here for extra emphasis:

"We compare our model results only to the central segment of the South American subduction zone, not its northern and southern branches, because our 2D models only represent the central segment of a wide subduction zone like South America."

We also now state in the abstract of our revised manuscript that our modelling and our model-nature comparison focus on the centre of the South American subduction zone:

"A key to help solve those issues is through studying the subduction zone dynamics

with 2D buoyancy-driven numerical modelling that uses constrained independent variables in order to best approximate the dynamics of the real subduction system in its centre."

We furthermore add clarification to the objectives statement on lines 40-44:

"The objectives of this paper are twofold: (1) calibrate independent variables for use in future 3-D modelling by comparing model outcomes with a range of geophysical and kinematic data of the central segment of the subduction zone, and (2) parametrically investigate the effect of the changed independent variables to get generic quantitative insights into how they affect subduction dynamics at the centre of the subduction zone."

Comment Fig. 3 for example shows changes in trench and plate velocity with single lines, are these averages along the whole length of the trench, or are these taken along a specific section? If the latter is true, then which section is it? This is extremely important because, as the authors show in Fig. 2, there are large variations along trench. For instance, the slab is not flat everywhere, but there are only 2 regions where this is the case. Importantly, along section BB' the slab is not flat and this is where the positive anomaly in the lower mantle is higher (meaning that the slab pile is more clear, following the reasoning of the authors). In fact, the other two sections, where the slab is flat, do not show the same large anomaly in the lower mantle. How does this reconcile with the main conclusions and, more generally, with the philosophy of the study?

Response What may have brought additional confusion in the original manuscript is that in Fig. 2 we also plotted profiles A-A' and C-C', which are not located in the centre, and that we did not state in the figure caption of Fig. 3 that the velocities are calculated for the central segment of the subduction zone. We now add a statement to the caption of Fig. 3 that "the velocities were calculated for the central segment of the subduction zone", and we keep profiles A-A' and C-C' in Fig. 2 in order to demonstrate that the South American subduction zone is quite symmetrical with respect to its centre regarding the upper mantle slab geometry. What should be added, and may also have

brought confusion, is that the values of slab dip angle close to the surface that we used as a basis for the model-nature comparison are between 0 and 25 degrees because the slab dip has evolved with time. Therefore, the present-day values range between 8–25 degrees (profile B-B' on Fig. 2) but we extended the range down to 0 degrees since flat slab is known to have occurred for the central segment as well [e.g. Ramos and Folguera, 2009]. For this last part we will include text in the revised manuscript explaining why the range is from 0 to 25 degrees (in the text and in the caption of Fig. 8).

Comment In other words, this study tries to find the best 2D model that fits as many criteria as possible with the natural case, but in South America the criteria themselves are not all present in one single 2D section of the subduction zone. So, what is the point of finding a model that fits everything, when even in nature this is not the case? I used the slab dip and the slab pile in the lower mantle as examples, but also the other criteria (like the subducting plate velocity and the trench velocity) are not the same all along the trench.

Response As discussed in our responses above, we do not intend to model or fit all parameter values along the subduction zone, but rather only the values measured and calculated for the central segment. We hope that this aspect is now clarified.

Comment Another important point is that the conclusions of this study are based on the comparison between models and data according to 9 criteria. However, at the moment, these criteria are mostly qualitative. How is the progressive reduction of trench velocity computed? What is the 'acceptable' range/error of flat slab portion length to have a rank of 1 or 2? The same goes for all criteria: what the authors define 'somewhat comparable' and 'very comparable' is subjective. I suggest to add a table in which the values of these criteria are clearly stated both for the models and for the data the models are compared with. And then again, some models might better fit one section of the subduction zone, but others that do poorly in that section might be better at fitting another section. I am not sure what we can learn from this though.

Response We have now defined a set of quantitative rules in order to determine the fitting score for all models. For example, based on the subducting plate velocity, a model gets 2 points if the average value after slab penetration into the lower mantle is within the natural range and the maximum and minimum values are within 10 percent of the natural maximum and minimum. The model gets 1 point if the average value after slab penetration into the lower mantle is within the natural range and the maximum and minimum values are not within 10 percent of the natural maximum and minimum. Otherwise the model does not get a point. We performed a similar analysis on all parameters and the main results remain similar as in the original manuscript. We clearly explain all the quantitative rules that we have used for our ranking in the revised manuscript.

Comment About the flat slab. The authors state that there is no need to add external forces or have a buoyant body to have flat slab and that slab flatting can happen dynamically as a consequence of a progressive decrease of the slab dip (lines 438-443). Then one might wonder why are there only two portions of flat slab along the South America subduction zone. Why is the slab not flat along section BB' (Fig. 2)? Moreover, these models do not take into the presence of buoyant bodies in the subducting plate (which it cannot be denied it is the case in the Nazca plate with the ridges entering the subduction zone), so the question is how would the results and conclusions change if a more realistic subducting plate would be modelled?

Response As we noted above, a flat slab also occurred at the central segment ("Altiplano flat slab") in the geological past (40-32 Ma and 27-18 Ma) [Ramos and Folguera, 2009]. Some of our most recent (unpublished) models and models presented in Schellart [2020] indicate that the slab dip angle close to the surface strongly evolves with time and may display an episodic behaviour. The aim of this paper is not to model natural complexity by adding all the properties of the subducting plate (aseismic ridges). We rather propose to first scale parameters using a relatively simple geometric and rheological setup in order to bring and study complexity stepwise. The next step indeed will

be to investigate the effect of aseismic ridges with a 3D setup.

Comment 40-44: I find the objectives of the paper very vague, they could fit with any paper that presents a parametric study. I suggest to be more specific about which features of the South America subduction zone the authors are trying to explain/understand with this study. For example, in the first sentence of the introduction it is stated that this subduction zone is enigmatic because of the orogeny formed with an oceanic subduction, but this study is not really addressing this issue. Instead, paragraph 2.5 is describing the features that are then compared with the results. To me, it makes more sense to move this paragraph in the introduction, but the authors should also add an explanation on what it is about these specific features that is not yet understood and what are the main research questions related to these features they are trying to answer.

Response As we explain above, we are trying to explain and understand the central segment of the South American subduction zone. Our view may moreover differ from the reviewer's opinion about parametric studies and their usefulness. We think that a parametric study can represent a goal on its own. Our study, in particular, represents an attempt to 1) scale independent variables using a model-nature comparison in which the model evolves self-consistently and many natural dependent variables are compared with the model outcomes. This is an important goal since it allows geodynamic modellers to re-use the scaled parameters in future modelling studies. 2) In our study, we moreover investigate the effect of the tested parameters on subduction dynamics, kinematics, slab geometry and overriding plate deformation. This is already interesting on its own since it could provide a better understanding of the rheological effects (upper mantle rheology, subduction interface strength, slab thermal weakening) on subduction dynamics. One particular point that is difficult to reproduce with geodynamic models is the fast subducting plate velocity observed for the Nazca-Farallon plate, which therefore justifies our choice of investigated parameters since they all can affect this velocity. Our conclusions moreover show that it is important to include thermal weakening of the slab to reach a fast subducting plate motion, which therefore has

implications for future modelling studies. We agree with the reviewer that our study is not really addressing the topic of orogenesis at a subduction zone, so in our revised manuscript the first sentence of the introduction has been changed.

Comment 243: from Fig. 3a it seems more that the gentle decrease in vT (between 2.5 and 2 cm/yr) is only until about 12 Ma, then there is a clear step to 1.5 cm/yr and the trench velocity remains more or less constant. Given that this is one of the thing that decides the final ranking of the models, how does this affect the results? Again, having a table with quantitative values for each criterion would help.

Response The trench retreat velocity was estimated using motion of the South American plate. Thus, the landward-directed deformation of the trench due to shortening in the Andes is not included in our calculation. However, this would result in further decrease in trench retreat rate, notably in the last ∼40 Myrs (e.g. Fig. 3 in Faccenna, 2017). We have clarified this point in the text of section 2.5, as follows: "Because our calculation does not consider landward-directed deformation of the trench due to shortening in the Andes, the progressive decrease in vT could be more significant (Faccenna et al., 2017)."

Comment The amplitude of oscillation in vSP is also something that the authors look at in the models, however this is not described in paragraph 2.5 at all, but it starts to be mentioned only in the results and discussion. Describe the natural range.

Response We do describe this in paragraph 2.5, namely on L241: "it fluctuates between ∼6 and ∼10 cm/yr in the past ∼45 Myr". To emphasize this better in the revised manuscript, the text "it fluctuates" is replaced with "the amplitude of oscillation in Vsp fluctuates".

Comment The viscosity jump between upper and lower mantle has a major control on the slab bending, piling and flattening. And slab piling and flattening is the main focus of this study. However the viscosity jump is not a parameter that is investigated. 100 is a commonly used value, but it is also an end-member. Often, other numerical studies

use 30 as a viscosity jump. How do the authors think a lower viscosity jump would affect their results? I would like to see a model with a lower viscosity jump and see the effect on slab piling and flattening.

Response We have also conducted models with different lower mantle viscosities and densities, but have decided to present the results of these models in another paper focusing on lower mantle properties. Indeed, we found that all the models with a lower viscosity for the lower mantle produce less slab folding and give a reduced fitting score, which is why we kept only models with a lower mantle viscosity of 100 in this paper. We have added a brief discussion to section 4.5 where we discuss the fitting scores of our models, to explain that using a reduced lower mantle viscosity will reduce the fitting score for all models.

Comment Other minor points 128: Does the assumption of a neutrally buoyant OP affect the overriding plate deformation?

Response We do not know but we expect that the effect is moderate based on a comparison with published models that include a positively buoyant overriding plate [Schellart, 2020]. In any case, this would not change the conclusions of the paper since all models have been run with the same overriding plate.

Comment The overriding plate is 60 km thick for about 1000 km from trench. How does this compare to South America?

Response It actually fits quite well the present-day thickness estimated geophysically [e.g. Heit et al., 2007]. It moreover considers the forearc and backarc thickness determined by Curie and Hyndman [2006], which suggests that these forearc and backarc are/were prevalent features in ocean-continent subduction zones of the Pacific domain (L162-164 of original manuscript).

Comment 196-198: is it only the viscosity of the slab that is affected by the thermal weakening? Or is it also the viscosity of the mantle? One of the reasons for this

question is also because in Fig. 12 there are 'blue', thus very weak, regions around the slab in the lower mantle. It seems to be a consequence of numerical instability, is it?

Response Yes, only the viscosity of the slab is affected by thermal weakening (L195-198 of original manuscript). In Fig. 12, the blue regions of the lower mantle are actually zones of the slab that are stretched because of the reduced viscosity. Thus, they are not a consequence of numerical instability.

Comment 291-292: the subducting plate does not seem to be entirely consumed in Fig. 6f, why does the model stop?

Response The model does not stop but continues for several Myrs. We did not show these late subduction stages on the figures because we thought that there are already enough figure panels in this paper and the late subduction stages display characteristics that are similar to the last figure panel in the manuscript.

Comment Fig. 6 only show two of the models with different A, I suggest to show the dynamics of the other two models (or at least a final snapshot) in the supplementary material.

Response Yes, Fig. 6 shows the models with the two extremes of the A value. However, Fig. 5 already shows the reference model with an intermediate A value. We can add the model with the other A value in the supplementary material if needed.

Comment 374: "the higher E', the quicker the flat slab is attained". Or is it simply because subduction is faster, but the amount of convergence is the same?

Response Yes, in those models, the higher slab thermal weakening, the faster subduction and thus the quicker the flat slab is attained.

Comment 729: then why is the natural range in figures (blue area) only going between 6 and 10 cm/yr?

Response The natural range for the subducting plate velocity will be updated to 3–10 cm/yr. This will not affect the results and conclusions of our paper.

References

Curie, C. A. and Hyndman, R. D.: The thermal structure of subduction zone back arcs, J. Geophys. Res. Solid Earth, 111(8), 1–22, doi:10.1029/2005JB004024, 2006. Faccenna, C., Oncken, O., Holt, A. F. and Becker, T. W.: Initiation of the Andean orogeny by lower mantle subduction, Earth Planet. Sci. Lett., 463, 189–201, doi:10.1016/j.epsl.2017.01.041, 2017. Heit, B., Sodoudi, F., Yuan, X., Bianchi, M. and Kind, R.: An S receiver function analysis of the lithospheric structure in South America, Geophys. Res. Lett., 34(14), L14307, doi:10.1029/2007GL030317, 2007. Ramos, V. A., and Folguera, A., Andean flat-slab subduction through time, in Ancient orogens and modern analogues, edited by J. B. Murphy, J. D. Keppie and A. J. Hynes, pp31-54, The Geological Society of London, Bath, doi:10.1144/SP327.3, 2009. Schellart, W. P.: Andean mountain building and magmatic arc migration driven by subduction-induced whole mantle flow, Nat. Commun., 8(1), 1–13, doi:10.1038/s41467-017-01847-z, 2017. Schellart, W. P.: Control of Subduction Zone Age and Size on Flat Slab Subduction, Front. Earth Sci., 8, 1–18, doi:10.3389/feart.2020.00026, 2020. Schellart, W. P., Freeman, J., Stegman, D. R., Moresi, L. and May, D.: Evolution and diversity of subduction zones controlled by slab width, Nature, 446(7133), 308–311, doi:10.1038/nature05615, 2007.

Please also note the supplement to this comment:
https://se.copernicus.org/preprints/se-2020-134/se-2020-134-AC1-supplement.pdf
* * *

---

## Author Comment (AC2) · 23 Sep 2020

Comment The authors present a parametric modeling study that aims to determine the physical subduction parameters associated with South American subduction. Overall, the figures are text are clear, the descriptions of the model behaviors are very well done, and the topic is interesting. However, I think there are some major issues with the study relating to i), the thermal weakening rheology adopted in a set of models, and ii), the overall study design. For the study to achieve its goals, I therefore think significant work is needed to re-think important aspects of the paper.

Response We thank Anonymous Referee #2 for his review work on our manuscript and for providing a critical analysis that will allow to improve and clarify the manuscript. Two main criticisms are raised by the reviewer.

1) The first point raised concerns the rheological implementation of our temperature-dependent viscosity to model thermal slab weakening. This is not a major concern since the issue may partly stem from a lack of clarity regarding our description of the distribution of different viscous rheologies and rheological laws in our layered setup (compositional and temperature-dependent), as well as from an omission in Eq. 14. Thus, this is partly a matter of clarification that we will address in a revised version of the manuscript (see detailed response below). In addition, the reviewer's comment may partly originate from his/her own preference in the way the rheological setup is built. However, in our opinion, there is not just one correct way to build a model rheologically and we think that our approach, in addition to the one suggested by the reviewer, is also valid and it has led to consistent published subduction models by other groups (see detailed response below).

2) The second criticism relates to the fact that we use a two-dimensional model setup to study a three-dimensional subduction system of which the investigated parameters vary along the trench, thus in the third dimension that is not included in our models. This comment is similar to the first main comment of Anonymous Referee #1. As discussed in our detailed response to the comments raised by the first reviewer, using a 2D model setup is not a major issue because it is appropriate to study the dynamics at the centre of wide subduction zones such as for South America. So in the manuscript we now put extra emphasis on the fact that we aim to model only the centre of the South American subduction zone (Bolivian orocline region), and not the segments to the north and south. This point was already mentioned in the methods section of the original manuscript (L98-101). In the revised manuscript we will make it clearer by adding text in the abstract, introduction, methods and captions of Fig. 2 and 3 (see detailed responses below and also in the response to the comments from Anonymous

Referee #1).

Comment Specific comments My first concern relates to the exploration of a thermal weakening rheology. This is important as these are the most successful suite of models. Ultimately, I am not sure your rheological implementation corresponds to what you intend. In the rheology section, you first describe a standard T-dependent Arhenius flow law (Eq. 12) and mention that the lithospheric layers have variable viscosities controlled by composition (lines ~185). In a compositional model, you would typically neglect the temperature dependence of the viscosity in the lithosphere (with the view that the compositional strengthening is mimicking this). Because you have both lithospheric composition and cold temperature, I am not sure how you dealt with this? (But I presume you did sufficiently as the viscosity field does look reasonable in the first non-thermal weakening figures). More importantly, "thermal weakening" is then introduced as another T-dependent viscosity (Eq. 14). This expression is just a linearized version of Eq. 12 and so I am not sure: why it is referred to as thermal weakening, how it is combined with the other two viscosities (compositional and Arrhenius), and what process it describes.

Response This issue may have been raised partly due to a lack of clarity in the original manuscript and an omission in Eq. 14. The T-dependent Arhenius flow law of Eq. 12 is used only for the upper mantle to approximate the dislocation creep deformation regime and not for the lithosphere layers, for which we used homogeneous viscosity values (and a visco-plastic rheology for the top layer of the subducting plate). The only exception is for those models that model thermal slab weakening, in which we indeed included a simplified version of Eq. 12, giving Eq. 14. This was clearly stated in the original manuscript (L175-180 and L195-197):

"Apart from the subducting plate top layer, all other plate layers and the lower mantle are Newtonian whereas for the upper mantle both a Newtonian and power-law rheology were tested (Table 2). In nature, mantle rocks deform by dislocation and diffusion creep following an Arrhenius flow law with temperature and pressure dependence (Karato

and Wu, 1993; Hirth and Kohlstedt, 2003). For dislocation creep, we computed a dynamic effective viscosity following the Arrhenius flow law but neglecting the effect of pressure (van Keken et al., 2008)". "In some models, thermal weakening of the slab is simulated using a non-dimensional temperature-dependent viscosity that follows a linearized Arrhenius flow law (Ratcliff and Schubert, 1996; Zhong et al., 2000)".

The linearized version of the Arrhenius flow law (Eq. 14) represents one way to use a dimensionless parameter (the dimensionless activation energy coefficient E' that controls the magnitude of the thermal slab weakening) in order to constrain this parameter and re-use it in future modelling. However, what may have confused the reviewer is that we omitted to write the equation as a function of the initially prescribed compositional viscosity for the slab layers. Thus, we have added a parameter (ðÌIJĆC_SP) in our methods to describe the compositional viscosity of the lithospheric layers, and we have added ðÌIJĆC_SP in Eq. 14, so that the right side of the equation becomes multiplied by ðÌIJĆC_SP. Therefore, the thermal dependence is applied on the initially prescribed compositional viscosity of the subducting plate layers only. To further clarify this, we now state after Eq. 14: "Using equation 14 allows us to model warming-induced viscosity reduction on the compositionally defined subducting plate layers".

This issue may have also partly raised because of the reviewer's preference to use a composite rheological law applied to a single material, thus without using a compositional layering, as in Garel et al. [2014]. However, as described above, we used an alternate approach in which both a compositional layering and rheological laws are used. This combined approach has been used and published by other groups and it has proven efficient to model subduction dynamics [e.g. Arredondo and Billen, 2016; Holt et al., 2015a]. So we think that using this combined approach is justified.

Comment Looking at the figures, it has the non-intuitive effect of lowering viscosities in the cold temperature lithosphere, particularly in the lower mantle (which doesn't agree with Eq. 14). What does this correspond to physically? (The studies that you cite – Ratcliff and Schubert (1996) and Zhong et al. (2000) - just use this flow law to

approximate a regular temperature dependent viscosity which, in disagreement with your models, produces strengthening in cold regions).

Response No, our thermal slab weakening actually decreases viscosity in warm regions of the slab. Thus, it is consistent with the equation used in Ratcliff and Schubert [1996] and Zhong et al. [2000]. The only difference with these earlier works is that our models do not show rheological strengthening since the slab does not go into cold regions.

Comment My other concern relates to the design of the study. The goal is the determination of geodynamic parameters for future 3-D modeling from the current 2-D models. However, the parameters chosen for exploration are not well justified. Why do you focus on these three parameters? If you are just trying to nail down a geodynamic reference setup, there are many other parameters that could significantly influence subduction and are just as uncertain: e.g. slab strength and rheology, lower mantle strength, oceanic plate density (e.g. plateaus), upper plate rheology. I think you need to either explore a larger range of parameters or provide more robust justification for your choices.

Response One particular point that is difficult to reproduce with geodynamic models is the fast subducting plate velocity observed for the Nazca-Farallon plate. Thus, we have decided to focus on this set of parameters (upper mantle rheology, subduction interface strength and slab thermal weakening) because we believe that they can strongly affect the subducting plate velocity, and they have proven to do so as shown in our results. We will include text in the revised manuscript explaining more clearly why we chose this set of parameters. Note that slab strength and rheology are actually treated indirectly using the slab thermal weakening. We also conducted models varying lower mantle properties but they will be presented in another paper since the work contains as much material as for this paper. Indeed, we found that all the models with a lower viscosity for the lower mantle produce less slab folding and give a reduced fitting score, which is why we kept only models with a lower mantle viscosity of 100 in this paper. We have added

a brief discussion to section 4.5 where we discuss the fitting scores of our models, to explain that using a reduced lower mantle viscosity will reduce the fitting score for all models. Then, we would like to add that we found a best-fitting model without testing the rheology of the overriding plate. This means that the best-fitting model can be used as such in order to include more complexity (third dimension, aseismic ridges) without changing the overriding plate properties because they already provide a good fit. However, we acknowledge that it would also be interesting to investigate how overriding plate properties affect subduction dynamics. We note that overriding plate properties have been studied considerably in subduction modelling studies [e.g. Holt et al., 2015a; Holt et al., 2015b] and we thus decided to focus on other parameters that we thought could affect more the subducting plate velocity.

Comment Second, trying to find a 2-D reference model for a very 3-D subduction system that exhibits strong along-strike variation (e.g. flat slab vs. no flat slab) seems challenging. I acknowledge you need a starting point for future 3-D models, and so it's probably worthwhile, but I think this produces extra concerns that should be addressed. For instance, one of the fit criteria is flat slab subduction. What about the regions that don't have flat slab subduction? Is a SAmerica reference model that produces flat subduction appropriate? (Especially given proposed links to buoyant oceanic plateaus in this region.) Also, at what latitude are the plate velocities extracted (Fig. 3) for comparison with models? Are they representative of the whole margin? For dips, you consider the along-strike range which seems very sensible. Perhaps a similar approach is also needed for the plate velocities?

Response This comment is similar to the first main comment raised by the first reviewer. So we would like to emphasize here as well that with our models and our model-nature comparison, we only focus on the central segment of the South American subduction zone (lines 98-101 in the Methods of our original manuscript). Indeed, our 2D model approach dictates that it is only (approximately) applicable to the central segment of a wide (and symmetrical) subduction zone [e.g. Schellart et al., 2007; Schellart, 2020],

of which the South American subduction zone is the best present-day example on the Earth. In any case, we realize now that we could have stated this more explicitly in our manuscript. Thus, to avoid any potential confusion in the future, we now state more explicitly that our model-nature comparison is only applicable to the central segment (Bolivian orocline region) of the South American subduction zone.

To accommodate the comment from the reviewer we will revise the first sentence of the Methods section such that our revised manuscript reads:

"The regional models were designed to conduct a parametric investigation on the effect of upper mantle rheology (linearly or non-linearly viscous), subduction interface yield stress $\sigma_y$ and slab thermal weakening on the subduction dynamics of the central segment (Bolivian orocline region) of the South American subduction zone over a long timescale ($\sim$60-200 Myr) and large spatial dimensions (11600 km laterally and 2900 km vertically)."

We have also modified the sentence on lines 98-101 in the original manuscript by adding several references that support our claim:

"The 2-D approach is a reasonable approximation considering that we simulate the subduction process at the centre of a very wide subduction system where toroidal mantle flow is minimal [Schellart et al., 2007; Schellart, 2017] and slab geometry and plate kinematics are very similar as in 3-D subduction models at the centre of the subduction system [Schellart, 2020]."

We have also added another sentence here for extra emphasis:

"We compare our model results only to the central segment of the South American subduction zone, not its northern and southern branches, because our 2D models only represent the central segment of a wide subduction zone like South America."

We also now state in the abstract of our revised manuscript that our modelling and our model-nature comparison focus on the centre of the South American subduction zone:

"A key to help solve those issues is through studying the subduction zone dynamics with 2D buoyancy-driven numerical modelling that uses constrained independent variables in order to best approximate the dynamics of the real subduction system in its centre."

We furthermore add clarification to the objectives statement on lines 40-44:

"The objectives of this paper are twofold: (1) calibrate independent variables for use in future 3-D modelling by comparing model outcomes with a range of geophysical and kinematic data of the central segment of the subduction zone, and (2) parametrically investigate the effect of the changed independent variables to get generic quantitative insights into how they affect subduction dynamics at the centre of the subduction zone."

We acknowledge that we did not state in the figure caption of Fig. 3 that the velocities were calculated for the central segment of the subduction zone. We thus now add a statement to the caption of Fig. 3 that "the velocities were calculated for the central segment of the subduction zone".

Comment Other comments 75-84: Relates to my first main comment but many studies consider temperature dependent slab viscosities (e.g. Garel et al., 2014, G-cubed) but, importantly, not in combination with a compositional slab viscosity.

Response In our view using a combined approach including compositional layers and rheological laws is not an issue. Other groups have published models using this approach [e.g. Arredondo and Billen, 2016; Holt et al., 2015a] and our models produce consistent results likewise.

Comment 100-105: If you are quoting a run times then you should also state the size of the model (e.g. total number of elements, degrees of freedom).

Response This is described at line 199 of the original manuscript. This information will also be indicated along with the run times in a revised version of the manuscript.

Comment Eq. 5: Is compositional buoyancy just applied to the upper plate and not the

subducting plate? I could not figure this out (also Lines 165-170).

Response The compositional buoyancy equals to zero in the subducting plate, so that we consider only the buoyancy as controlled by temperature. We have added text to clarify this point in a revised version of our manuscript.

Comment 240-250: You ignore the OP shortening component of vt. Fair enough, but could this be why you get a progressive decrease in vt (Fig. 3b)? If so, is it appropriate to match this vt trend with models that don't have significant OP deformation?

Response We indeed ignore overriding plate shortening and its associated effect on trench motion when estimating the trench retreat rate in nature using motion of the South American plate. However, the reduction in Vt would actually be more pronounced if we would include overriding plate shortening, as shown in Faccenna et al. [2017].

Comment 503: Factor 100 lower mantle viscosity increase is probably reasonable, but on the high end. Did you test a reduced value?

Response Yes, we did test lower values but as we explained above these models all gave a worse fit (lower fitting score) and so we plan to present those models in another paper focusing on lower mantle properties.

Comment 532: According to what reasoning are these crustal yield stress values reasonable? Refs?

Response According to our results that suggest that values between ∼14–21 MPa produce the most consistent model outcomes and also according to earlier studies as we discussed at lines 651-658 of our original manuscript (e.g. 14-16 MPa [Seno, 2009], ∼10 MPa [Wang et al., 1995; Gutscher and Peacock, 2003], 15-30 MPa [Zhong and Gurnis, 1994], < 35 MPa [Duarte et al. 2015]). We have added text to clarify this in a revised version of our manuscript.

Comment 644: Ra # increased by reducing mantle viscosity? Or increasing slab/plate density?

Response Ra was increased by increasing the subducting plate density. We have added text to describe this in a revised version of the manuscript.

Comment 673: What is this Gilbert paper? Not in reference list. If similar to the Cerpa work, they do not solve for the viscous mantle but parameterize it using edge forces on the slab. So not really fair to call these infinite slab-mantle viscosity ratio models.

Response The reference is Gibert et al. [2012] and it is in the reference list. We have changed "infinite slab-mantle viscosity ratio models" to "high slab-mantle viscosity ratio models" since they used a viscosity ratio that is comparable to high end-members used in analogue models (6000–15000), which is thus ∼10–55 times higher than the ratio we used.

Comment Overall, the discussion is long winded and, in places, repetitive. Further effort to consolidate it around the main points would really improve readability!

Response Further effort will be made to make the discussion more concise in a revised version of the manuscript.

References

Arredondo, K. M. and Billen, M. I.: The effects of phase transitions and compositional layering in two-dimensional kinematic models of subduction, J. Geodyn., doi:10.1016/j.jog.2016.05.009, 2016. Duarte, J. C., Schellart, W. P. and Cruden, A. R.: How weak is the subduction zone interface?, Geophys. Res. Lett., 42(8), 2664–2673, doi:10.1002/2014GL062876, 2015. Faccenna, C., Oncken, O., Holt, A. F. and Becker, T. W.: Initiation of the Andean orogeny by lower mantle subduction, Earth Planet. Sci. Lett., 463, 189–201, doi:10.1016/j.epsl.2017.01.041, 2017. Garel, F., Goes, S., Davies, D. R., Davies, J. H., Kramer, S. C. and Wilson, C. R.: Interaction of subducted slabs with the mantle transition-zone: A regime diagram from 2-D thermo-mechanical models with a mobile trench and an overriding plate, Geochemistry, Geophys. Geosystems, 15(5), 1739–1765, doi:10.1002/2014GC005257, 2014.

Gibert, G., Gerbault, M., Hassani, R. and Tric, E.: Dependency of slab geometry on absolute velocities and conditions for cyclicity: Insights from numerical modelling, Geophys. J. Int., 189(2), 747–760, doi:10.1111/j.1365-246X.2012.05426.x, 2012. Gutscher, M. -A., and Peacock, S. M.: Thermal models of flat subduction and the rupture zone of great subduction earthquakes, J. Geophys. Res., 108(B1), doi:10.1029/2001JB000787, 2003. Holt, A. F., Becker, T. W. and Buffett, B. A.: Trench migration and overriding plate stress in dynamic subduction models, Geophys. J. Int., 201(1), 172–192, doi:10.1093/gji/ggv011, 2015a. Holt, A. F., Buffett, B. A. and Becker, T. W.: Overriding plate thickness control on subducting plate curvature. Geophysical Research Letters, 42(10), 3802-3810, 2015b. Ratcliff, J. T., Schubert, G. and Zebib, A.: Steady tetrahedral and cubic patterns of spherical shell convection with temperature- dependent viscosity, J. Geophys. Res. Solid Earth, 101(B11), 25473–25484, 1996. Schellart, W. P.: Andean mountain building and magmatic arc migration driven by subduction-induced whole mantle flow, Nat. Commun., 8(1), 1–13, doi:10.1038/s41467-017-01847-z, 2017. Schellart, W. P.: Control of Subduction Zone Age and Size on Flat Slab Subduction, Front. Earth Sci., 8, 1–18, doi:10.3389/feart.2020.00026, 2020. Schellart, W. P., Freeman, J., Stegman, D. R., Moresi, L. and May, D.: Evolution and diversity of subduction zones controlled by slab width, Nature, 446(7133), 308–311, doi:10.1038/nature05615, 2007. Seno, T.: Determination of the pore fluid pressure ratio at seismogenic megathrusts in subduction zones: Implications for strength of asperities and Andean-type mountain building, J. Geophys. Res. Solid Earth, 114(5), 1–25, doi:10.1029/2008JB005889, 2009. Wang, K., Mulder, T., Rogers, G. C. and Hyndman, R. D.: Case for very low coupling stress on the Cascadia subduction fault, J. Geophys. Res., 100(B7), doi:10.1029/95jb00516, 1995. Zhong, S., and Gurnis, M.: Controls on trench topography from dynamic models of subducted slabs, J. Geophys. Res., 99(B8), 15,683–15,695, doi:10.1029/94JB00809, 1994. Zhong, S., Zuber, M. T., Moresi, L. and Gurnis, M.: Role of temperature-dependent viscosity and surface plates in spherical shell models of mantle convection, J. Geophys. Res. Solid Earth, 105(B5),

11063–11082, doi:10.1029/2000jb900003, 2000.

Please also note the supplement to this comment:
https://se.copernicus.org/preprints/se-2020-134/se-2020-134-AC2-supplement.pdf